# A dual-reporter system for investigating and optimizing protein translation and folding in *E. coli*

Ariane Zutz[1], Louise Hamborg [1,2], Lasse Ebdrup Pedersen [1], Maher M. Kassem[2], Elena Papaleo[2], Anna Koza[1], Markus J. Herrgård [1], Sheila Ingemann Jensen [1], Kaare Teilum [2], Kresten Lindorff-Larsen [2] & Alex Toftgaard Nielsen [1]✉

Strategies for investigating and optimizing the expression and folding of proteins for bio-technological and pharmaceutical purposes are in high demand. Here, we describe a dual-reporter biosensor system that simultaneously assesses in vivo protein translation and protein folding, thereby enabling rapid screening of mutant libraries. We have validated the dual-reporter system on five different proteins and find an excellent correlation between reporter signals and the levels of protein expression and solubility of the proteins. We further demonstrate the applicability of the dual-reporter system as a screening assay for deep mutational scanning experiments. The system enables high throughput selection of protein variants with high expression levels and altered protein stability. Next generation sequencing analysis of the resulting libraries of protein variants show a good correlation between computationally predicted and experimentally determined protein stabilities. We furthermore show that the mutational experimental data obtained using this system may be useful for protein structure calculations.

[1] The Novo Nordisk Foundation Center for Biosustainability, Technical University of Denmark, Kemitorvet 220, 2800 Kgs, Lyngby, Denmark. [2] Structural Biology and NMR Laboratory, Department of Biology, University of Copenhagen, Ole Maaloes Vej 5, 2200 Copenhagen N, Denmark. ✉email: atn@biosustain.dtu.dk

Expression of heterologous proteins is essential for a number of purposes including functional and structural characterization, as well as for industrial production of enzymes and biochemicals through metabolic pathway engineering. However, heterologous expression of recombinant proteins in bacteria such as E. coli, a valued host for industrial expression of a wide range of peptides and proteins, often results in misfolding, aggregation and degradation. It is therefore of significant importance to be able to efficiently modify the protein coding sequence in a way that will enable more efficient folding and expression.

Several strategies for improving expression and folding of heterologous proteins are known, including for example screening and optimization of environmental factors such as host strain, growth medium and temperature, induction parameters and co-expression of folding chaperones[1]. Other strategies involve the use of protein affinity and solubility tags, which are short peptide or protein tags fused to the N- or C-terminus of a protein. Solubility tags function as folding scaffolds thereby helping to improve translation and folding of proteins with poor folding properties[2]. A small affinity tag is less likely to interfere with the three-dimensional structure of the protein[3], and it has the advantage that it can be used for affinity purification and for detection and quantification by Western blotting.

Another strategy for improving protein expression and folding involves optimization of the expression plasmid and gene of interest. The natural variation in codon usage often reflects changes in translation speed needed for correct co-translational protein folding[4–6]. Changes in codon usage or expression host may lead to changes in the translation rate, cause misincorporation of amino acid residues and truncations of the protein due to premature termination of translation. Structured parts of a protein may demand slower translation to enable co-translational folding, while more unstructured parts allow for more rapid translation[4,6–9]. Other factors to optimize include the choice of promoter, different mRNA secondary structures, optimal open reading frames (ORFs), and avoidance of certain amino acid residues in the N- or C-terminal of the protein as they can be susceptible for proteolysis or prevent initiation of the translation process[10,11]. Changing only one variable may not have the desired effect as many of these factors are linked and have a synergistic effect, thus emphasizing that the optimization process is not a straightforward task[12,13]. Furthermore, hydrophobic parts of the polypeptide chain are more prone to aggregation[14], and truncation of unstructured hydrophobic parts of the protein may thus improve protein folding. In addition, computational methods can be used to predict protein variants with optimized improved folding and expression properties[15].

A more efficacious way to improve protein expression would be to screen large random mutant libraries for variants of a protein with optimized folding. However, generation of random mutant libraries often results in frequent frame-shift mutations and stop codons. When screening for mutants with improved folding, it is therefore necessary to exclude the large number of clones that no longer express the target protein or form aggregates. It would thus be desirable to screen simultaneously for folding and expression. Many such methods for the analysis of protein expression and folding require extraction of proteins from the production organism, separating the proteins into soluble (folded) and insoluble fractions, and analyzing these fractions using SDS-PAGE or dot-blot based technologies[16–18]. These time-consuming processes are not amenable for screening of larger libraries of production organisms or protein variants at the single-cell level. Several bacterial systems have been developed for testing and screening variants for expression or stability. Examples include fusion reporter proteins for assessing protein folding and solubility using fluorescence, enzymatic reactions, antibiotic resistance or ligand binding as reporters for the production of soluble and folded proteins[19–26]. However, no system enables simultaneous monitoring of translation and folding at the single cell level.

Although proteins are generally able to fold into their native conformation by themselves, most organisms have evolved mechanisms for controlling and aiding the process to prevent unproductive misfolding. Molecular chaperones are constitutively expressed and participate in de novo protein folding by stabilizing the nascent polypeptide chain on the ribosome, in protein trafficking and domain assembly, and assist in degradation of partially folded and aggregated proteins. Several bacterial chaperones are induced when misfolded proteins are expressed in the cell and their promotors may be used to drive stress induced heterologous protein expression[27]. Thus, chaperone promoters can be used to construct reporters for the presence of e.g., misfolded protein. Previous work has focused on coupling such promoters to the expression of luciferase[28] or beta-galactosidase[29], which both require chemical assays for assessing their activity. Several methods are available for monitoring the level of protein production, such as it has been demonstrated using a translation-coupling system in E. coli[30]. None of the current methods, however, are suited for high-throughput approaches with simultaneous but independent in vivo monitoring of both translation and protein folding.

Here, we demonstrate a functional dual-reporter system that enables single-cell monitoring of both protein translation levels and the occurrence of protein misfolding. The system can be used to analyze translation levels and folding properties of heterologously expressed proteins in E. coli. We demonstrate the use of the system for screening the expression levels of various proteins including the effect of different solubility tags. We further show how the system can be combined with fluorescence activated cell sorting (FACS) and next generation sequencing (NGS) in a deep mutational scanning experiment[31] for generating protein wide identification of mutations important for correct protein translation and folding. We find a resonable agreement between computationally calculated protein stability of mutant PARP1-BRCT proteins and experimental data. Furthermore, we show that the mutational experimental data obtained in this work can be used to select native-like structures from a large pool of structures highlighting the usefulness of such systems in protein structure calculation.

## Results

**Dual-reporter system**. To enable high throughput analysis, we have developed a dual-reporter system that simultaneously monitors protein translation and protein folding at the single cell level (Fig. 1a). The translation sensor consists of a translation-coupling cassette comprised of a strong secondary mRNA structure formed by a C-terminal hexa-histidine tag, a stop codon for the gene of interest and a ribosome binding site (RBS) for a downstream fluorescent reporter protein, mCherry. The cassette has been inserted into a modified pET22b plasmid containing the pBR322 origin of replication (ORI) (Fig. 1b). When the gene of interest is correctly translated the secondary mRNA structure will be unfolded by ribosomal helicase activity, and expose the RBS for the downstream reporter gene enabling RNA polymerase to continue transcription[30]. An untimely termination of the translation will hinder the ribosome reaching the position of the mCherry gene, thus preventing a fluorescent signal to emerge. Correct translation of the gene of interest results in the expression of mCherry being proportional to the expression of the protein of interest.

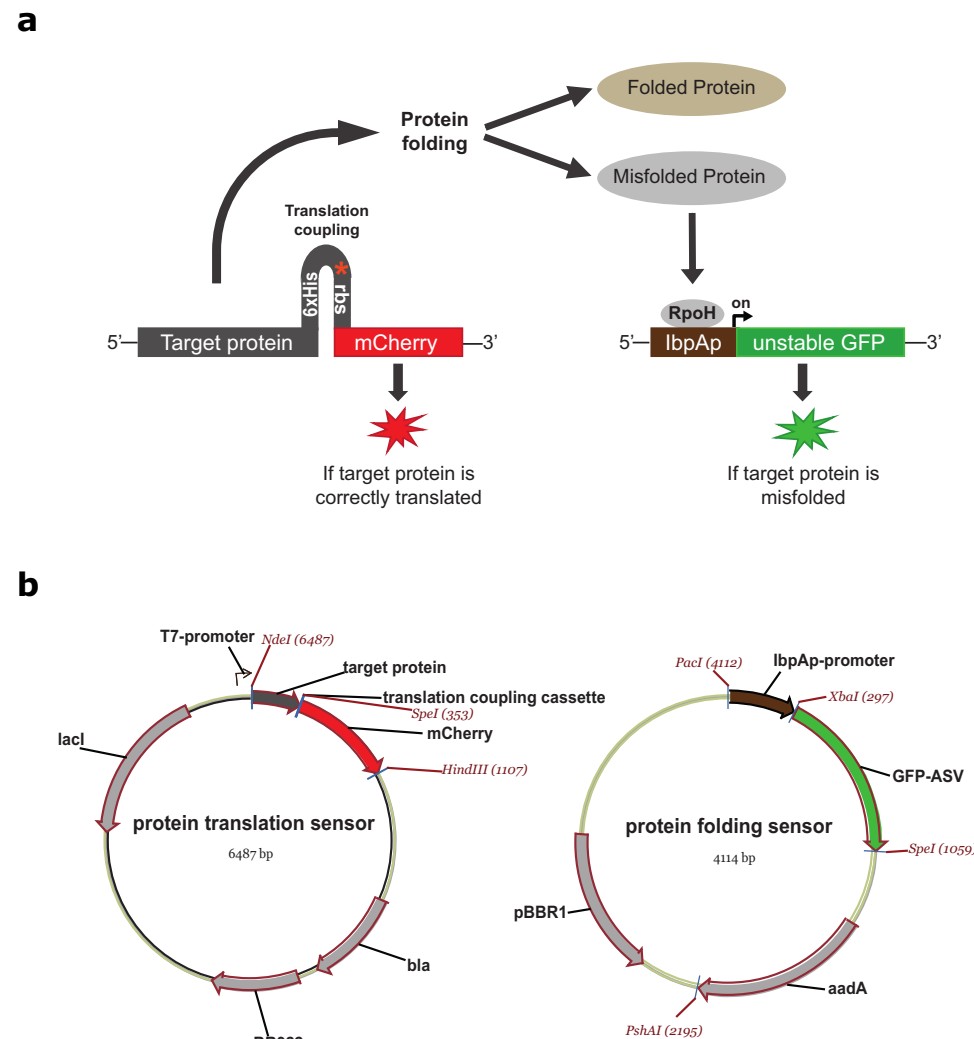

**Fig. 1 Schematic overview of a dual-reporter system with simultaneous monitoring of protein translation and protein folding at the single cell level.** **a** The translation sensor is comprised of the gene of interest translationally coupled to the reporter protein mCherry. When the target gene is correctly translated, the RNA polymerase unfolds the secondary structure and the mCherry gene is transcribed resulting in a red fluorescent signal. The synthesized polypeptide chain then either folds into a soluble protein conformation or it fails to fold, thereby typically forming protein aggregates that accumulate as inclusion bodies. Formation of inclusion bodies increases the cellular level of free RpoH (heat shock sigma-factor $\sigma^{32}$). RpoH binds to the lbpA promoter in the protein folding sensor, initiating the expression of an unstable GFP variant, GFP-ASV, yielding a green fluorescent signal. **b** Overview of the plasmids used for the protein translation and protein folding sensors.

The protein folding sensor is based on the naturally occurring heat shock response system in *E. coli*. Heat shock proteins (HSPs) are expressed to protect the cell when exposed to high temperatures or other forms of stress condition. HSPs are often molecular chaperones that bind the hydrophobic parts of partially unfolded proteins and assist in refolding and protection against degradation by proteases. In *E. coli*, the alternative sigma factor 32, called RpoH, controls the transcription of several cytoplasmic HSPs, including the small inclusion body HSPs, IbpA/B[32,33]. In an unstressed cell RpoH is bound to chaperone DnaK, but during stress RpoH will be released when DnaK binds unfolded protein, thus increasing the level of free RpoH in the cell. RpoH then binds to the core RNA polymerase forming a holoenzyme complex, which subsequently recognizes heat shock promoters and thus initiate a heat shock response[34]. In our protein folding sensor, the RpoH inducible lbpA promoter is inserted upstream of a GFP reporter gene in a modified pSEVA631 vector[35] with the medium-copy pBBR1 ORI (Fig. 1b). When the dual-reporter system is used, the formation of misfolded protein will initiate expression from the lbpA promoter resulting in the expression of GFP.

**Effect of plasmid backbone and GFP variant on signal distribution.** To test whether the copy number affects the lbpAp-GFP activity, we have analyzed the heat shock response from two vector backbones with the pBBR1 or the ColE1 ORI, respectively. The two ORIs were further tested in combination with two GFP-variants, GFP-mut3 and GFP-ASV. GFP-mut3 is a stable GFP-variant with a half-life estimated to more than 1 day, while a C-terminal degradation tag makes GFP-ASV susceptible to protease degradation and results in a shorter half-life of about 110 min[11].

Cell cultures in the exponential growth phase were exposed to 42 °C for 10 min to induce a cellular heat shock response. The heat shock response was followed by monitoring the GFP fluorescence for 2 hours (Fig. 2a). Immediately after the exposure to 42 °C, a rapid increase in GFP expressed from plasmids with

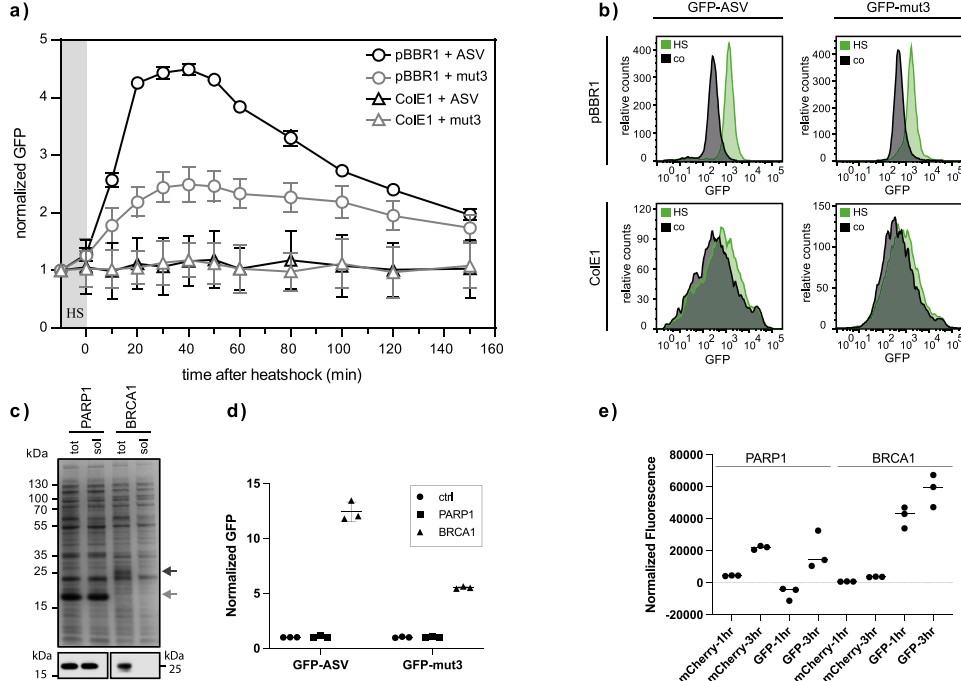

**Fig. 2 Optimization and test of the protein folding sensor plasmid to improve differentiation of heat shock response signals. a** Monitoring of the heat shock response signals of protein folding sensors (pSEVA441-IbpAp and pSEVA631(Sp)-lbpAp) with different origin of replications (ColE1 and pBBR1, respectively) and GFP variants (GFP-mut3 and GFP-ASV) after induction of the lbpA promoter. Changes in the GFP signal after induced heat shock (HS) are monitored using flow cytometry and the GFP signals in triplicates (average ± SD) are normalized to the respective background signal at each time-point. **b** FACS profiles for the GFP signals 60 min after induced heat shock for GFP-mut3 and GFP-ASV in plasmids with different origin of replications. Relative counts of GFP fluorescence intensities are shown from the analysis of 10,000 single cells. The heat shock induced (HS) GFP variants expressed from pBBR1 (pSEVA631(Sp)-lbpAp) shows well-defined and distinct peaks, which are easy to distinguish from the un-induced control plasmids (co). The GFP variants expressed from ColE1 (pSEVA441-IbpAp) resulted in very broad and not well-defined peaks making it difficult to distinguish between the heat shock induced plasmids and the control. **c** SDS-PAGE and immunoblot analysis of total (tot) protein yield and soluble protein (sol) after fractionated cell disruption of two human proteins, PARP1-BRCT and a truncated version of BRCA1-BRCT, shows high expression of a soluble PARP1-BRCT protein, and an insoluble BRCA1-BRCT protein. The shown data is representative of at least three repetitions. **d** Flow cytometry analysis 60 min after protein induction of the co-expression of PARP1-BRCT and BRCA1-BRCT with the pSEVA631(Sp)-lbpAp-GFP-ASV and pSEVA631(Sp)-lbpAp-GFP-mut3 plasmids. The soluble PARP1-BRCT does not initiate a heat shock response and results in a low green fluorescent signal, whereas the insoluble BRCA1-BRCT protein triggers the heat shock response causing a high green fluorescent signal. The pSEVA631(Sp)-lbpAp-GFP-ASV plasmid has an improved signal-to-noise ratio and is preferred over the pSEVA631(Sp)-lbpAp-GFP-mut3 plasmid. Data are presented as mean values ± standard deviation determined from three biologically independent experiments. **e** Plate reader analysis 1 h and 3 h after protein induction of the co-expression of PARP1-BRCT and BRCA1-BRCT with the pSEVA631(sp)-lbpAp-GFP-ASV plasmid in *E. coli* K-12 MG1655 (DE3). The soluble PARP1-BRCT initiates mCherry co-expression, but does not trigger the folding reporter signal, whereas the insoluble BRCA1-BRCT protein has less mCherry signal, but trigger the folding reporter response. Data are presented as mean values ± standard deviation based on three biologically independent experiments. Source data are provided as a Source Data file.

the pBBR1 ORI was observed reaching a maximal level after 20 min. The GFP signal was consistent with the expected change in RpoH synthesis rate observed during a heat shock induced response, where the formation of misfolded protein is known to initiate a spike in the RpoH synthesis rate, that slowly declines to a level higher than before the heat shock[34,36]. The heat shock promoter under the control of the ColE1 ORI plasmid did not give rise to a heat shock response signal.

Since differentiation between heat shock induced and un-induced GFP responses is crucial for the applicability of the folding sensor, single cell analysis was carried out. FACS profiles of the heat induced and un-induced ColE1 plasmids show broad overlapping peaks, indicating a leaky expression of GFP (Fig. 2b). The accumulation of GFP in the cell made it impossible to monitor a heat shock response signal different from the basal GFP level using the ColE1 based plasmids. In contrast, two sharp well-defined peaks were observed from the heat induced and un-induced pBBR1 plasmids with a 3–5-fold increase in the signal-to-noise ratio. The highly stable GFP-mut3 had a high basal fluorescence level causing a significant overlap between the

induced and control responses resulting in a low signal-to-noise ratio. The use of the short half-life of GFP-ASV resulted in a lower basal fluorescence level thereby giving higher signal-to-noise ratios, which enabled the distinction of the heat shock induced response from protein misfolding in single cells. To test the compatibility of the translation sensor and the protein folding sensor, we chose to analyze two human proteins with differences in expression levels and solubility in *E. coli*, PARP1-BRCT and BRCA1-BRCT. PARP1-BRCT and BRCA1-BRCT contain the BRCT domain of human Poly[ADP-ribose] polymerase 1 (PARP1) and human breast cancer 1, early onset (BRCA1), respectively. The BRCA1-BRCT construct was designed to promote misfolding by making a truncation of the folded BRCT domain. PARP1-BRCT was expressed in high yields as soluble protein in *E. coli* as shown by SDS-PAGE and Western blot analyses, while the BRCA1-BRCT domain was expressed as an insoluble protein in *E. coli* (Fig. 2c).

Folding of PARP1-BRCT and BRCA1-BRCT was further analyzed using pSEVA631(Sp)-IbpAp-GFP-ASV and pSE-VA631(Sp)-IbpAp-GFP-mut3 as protein folding sensors and

monitored by flow cytometry. As expected, the expression of the soluble PARP1-BRCT did not initiate a GFP response compared to the control carrying only an empty pET22b vector (Fig. 2d). Overexpression of the insoluble BRCA1-BRCT, however, promoted binding of RpoH to the IbpA promoter region of the folding sensor, resulting in a 5–10-fold increase in GFP-signal compared to PARP1-BRCT. As previously observed, the GFP-ASV variant yielded a higher fluorescent signal, and a better signal-to-noise ratio compared to GFP-mut3 (Fig. 2d). To confirm the applicability of the system in other *E. coli* strains, we further expressed the PARP1-BRCT and BRCA1-BRCT in *E. coli* K-12 MG1655 (DE3), and monitored the mCherry and GFP-ASV signals of the cultures in a plate reader after 1 h and 3 h, seeing a significantly higher GFP expression in the cells expressing BRCA1-BRCT (Fig. 2e). These results demonstrate the applicability of the protein folding sensor as a general tool for monitoring protein folding in vivo.

**Effect of protein solubility tags on translation and folding**. Overexpression of recombinant proteins in *E. coli* often results in misfolded proteins and the formation of insoluble aggregates. To enhance the solubility, fusion proteins are often linked to the N-terminus of proteins that aggregates during expression. To test the applicability of the dual-reporter system to monitor the effects of linking solubility tags, we investigated the fusion of two commonly used expression tags, the N-utilization A (NusA) and the small ubiquitin related modifier (SUMO), on the translation levels and solubility of four different model proteins. The proteins were chosen based on their different translation levels and tendency to form inclusion bodies (IB) when expressed in *E. coli*, and include PARP1-BRCT[37], a truncated variant of BRCA1-BRCT[38], the human cyclin-dependent kinase inhibitor, p19[39] and the viral oncogene E6 from human *papillomavirus type 16*[40]. Wild-type PARP1-BRCT, BRCA1-BRCT, E6, and p19 were cloned into the translation sensor with either NusA or SUMO linked to the N-terminus of the proteins. The translation and protein folding sensors were co-expressed at 30 °C. Translation and protein folding were monitored by flow cytometry, while SDS-PAGE and Western blot analyses were used to compare the levels of expressed and soluble protein. We observed a strong correlation between translation levels monitored by flow cytometry and the expression yield detected by Western blot (Fig. 3a). The N-terminally linked solubility tags did not have a large impact on the translation level of PARP1-BRCT, BRCA1-BRCT, and E6, which all expressed also without the tags, and the SUMO tag even decreased the expression level of BRCA1-BRCT. In contrast, under the given conditions wild type p19 did not express, however, fusion with either NusA or SUMO enabled expression, with NusA having a bigger effect.

The total amount of protein expressed, and the fraction of soluble protein were quantified by Western blotting and compared to the GFP fluorescence from the protein folding sensor monitored by flow cytometry (Fig. 3b). Expression of all PARP1-BRCT and the p19 variants resulted in background GFP fluorescence. In contrast BRCA1-BRCT and E6 expressed as insoluble aggregates and resulted in high GFP fluorescence. Different levels of GFP fluorescence were observed for BRCA1-BRCT and E6 although they both were expressed as insoluble protein. We also note that the fusion of the proteins to NusA or SUMO did not increase the amount of soluble protein, as also quantified by the folding sensor. The results demonstrate that it is possible to combine both biosensors to simultaneously investigate translation and proper folding of proteins in *E. coli*.

**Quantitative determination of protein stability and protein misfolding**. To test whether the folding sensor can be used to quantitatively measure protein stability and protein misfolding, six variants of the chymotrypsin inhibitor 2 (CI2) with different experimentally-determined thermodynamic stabilities ($\Delta G_U$)[41] were cloned into the translation sensor vector. CI2 is a serine protease inhibitor that has been extensively used as a model protein in protein folding and stability studies[41,42]. The protein variants were expressed at 30 °C using pSEVA631(Sp)-lbpAp-GFP-ASV as protein folding sensor. The GFP fluorescence monitored by flow cytometry was compared to the in vitro stability of the His-tagged proteins at 30 °C determined by global fitting of temperature and denaturant unfolding. The GFP fluorescence clearly changed with $\Delta G_U$, where more unstable proteins resulted in higher GFP signals (Fig. 4). These results show that the GFP fluorescence arising from the protein folding sensor can be used as a proxy for the in vitro stability of variants in a mutant library, by characterizing the stability for each variant as higher or lower than the starting point. These results also suggest that the dual-reporter system can be used for analysis and sorting of mutant libraries using flow cytometry based on translation levels, and that it may enable the selection of proteins with altered stability.

**Screening and sorting of protein wide mutant libraries**. Factors affecting proper folding of proteins can be investigated by random mutagenesis. However, stop-codons, frameshifts and indels will often be introduced in a randomly generated mutant library, which render it difficult and time-consuming to screen for new protein variants. We thus demonstrate the use of the dual-reporter system to screen for variants with either reduced or increased stability in a high-throughput mode. First, a randomly generated mutant library of pET22-PARP1-BRCT-mCherry was expressed with the protein folding sensor pSEVA631(Sp)-IbpAp-GFP-ASV to demonstrate the applicability of the dual-reporter system to screen for new variants with correct translation but altered stability. PARP1-BRCT is a stable protein resulting in a low GFP signal and a signal distribution corresponding to the control. We created a mutant library and used the folding sensor to analyze positions and variation important for proper cellular folding. The mutant library was prepared using the error-prone DNA polymerase Mutazyme II that provides a minimal mutational bias. By adjusting the amount of initial target DNA and the number of gene duplications, a mutation rate of 1–3 amino acid substitutions per protein was achieved. GFP and mCherry fluorescence was quantified by FACS 1 h after protein expression was induced by IPTG (Fig. 5a). PARP1-BRCT WT and the PARP1-BRCT mutant library showed a high translation level with well-defined mCherry signals that were distinct from the background fluorescence from an empty pET22b vector (Fig. 5a). Before sorting of the cells, similar GFP signals and distributions were obtained for PARP1-BRCT WT, the PARP1-BRCT mutant library, and the background control. The cell cultures were sorted using FACS for high mCherry signal (P1) alone and for both high mCherry signal (P1) and high GFP signal (P2). Gate 1 (P1) was defined as a mCherry signal higher than the control plasmid background, to ensure that only cells expressing correctly translated proteins were collected. Gate 2 (P2) was defined as the upper 1% of cells with the highest GFP fluorescence, in order to select for variants expressing proteins with decreased stability. The collected cells were grown and sorted again using the same criteria as in the initial sorting. The fluorescence from the final pools of sorted cells was analyzed by FACS 1 h and 3 h after induction of protein expression by IPTG

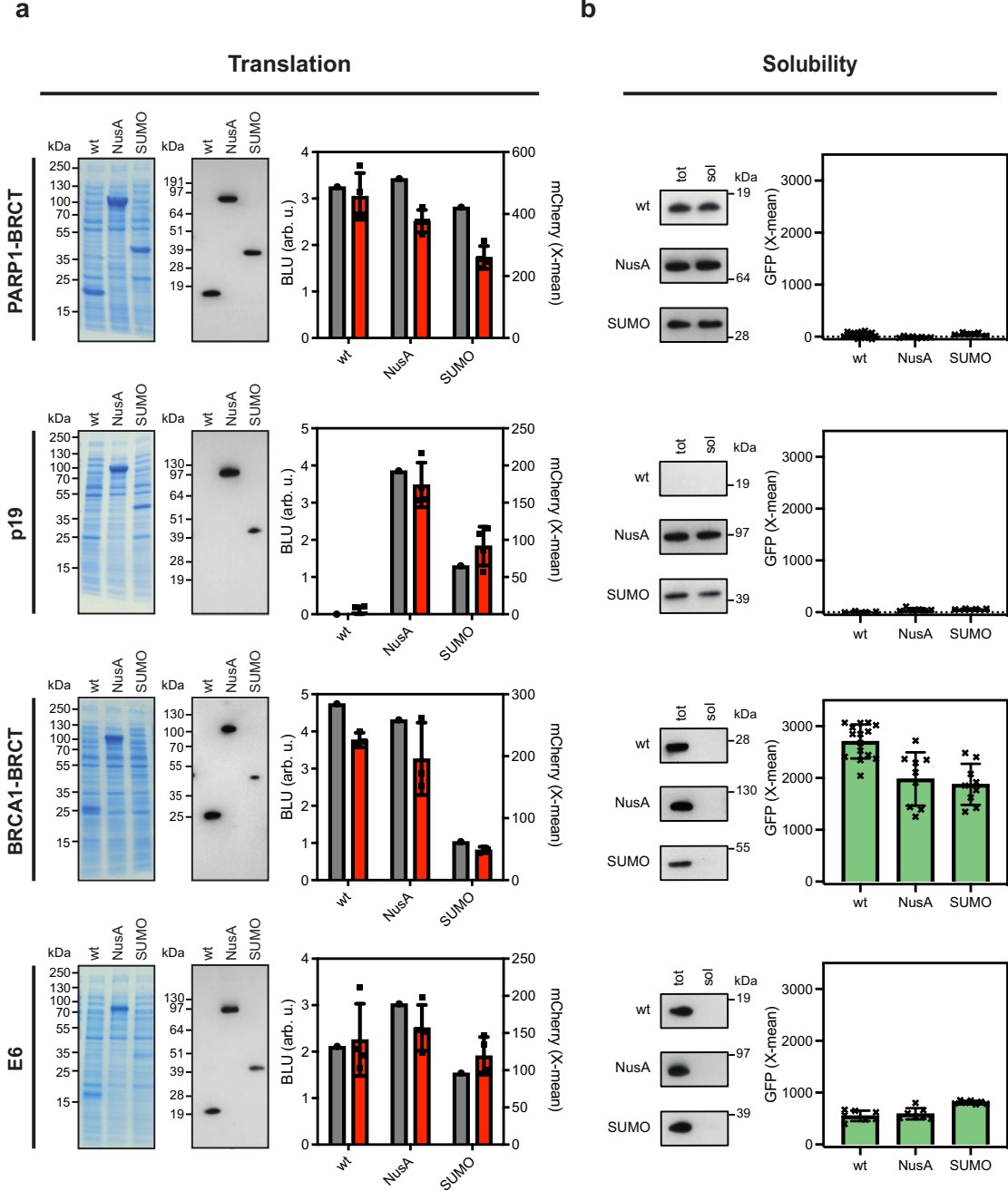

**Fig. 3 Validation of the dual protein translation and misfolding biosensor.** The solubility tags NusA and SUMO were fused to four proteins; PARP1-BRCT, p19, a truncated BRCA1-BRCT, and E6, with known propensities for misfolding. **a** The proteins were translationally coupled to the fluorescent protein mCherry to monitor the translation using FACS. Data are presented as mean values ± standard deviation for biologically independent samples analyzed for each plasmid combination ($n = 4$ for PARP1-BRCT, p19 and E6, and $n = 3$ for BRCA1-BRCT). Protein expression was analyzed by SDS-PAGE analysis and quantified from a single Western blot for each cell line (gray) (BLU = biochemical luminescence unit) and correlated to the mean mCherry fluorescence signal from the analysis of 10,000 cells (red). **b** Western blot analysis of total protein yield (tot) and soluble protein (sol) after fractionated cell disruption shown together with the quantified GFP response signal for insoluble protein. Data are presented as mean values ± standard deviation based on 4 biologically independent samples. Source data are provided as a Source Data file.

(Fig. 5b). The PARP1-BRCT library sorted for high mCherry signal (Lib. P1) represents correctly translated proteins, and show similar GFP intensities and signal distributions as the PARP1-BRCT WT. The PARP1-BRCT library sorted for both high mCherry signal and a high GFP signal (Lib. P2) shows a clear shift in GFP signal compared to PARP1-BRCT WT and the PARP1-BRCT library that was only sorted for correct translation (Lib. P1) (Fig. 5b). The shift in GFP signals indicates

that protein variants with impaired folding properties had been enriched. Furthermore, the PARP1-BRCT library sorted for high GFP fluorescence showed a small broadening of the GFP signal 3 h after induction due to continuous expression of GFP concurrently with PARP1-BRCT being expressed. Single cells were extracted from the sorted libraries and the PARP1-BRCT gene was amplified by PCR and prepared for DNA amplicon sequencing.

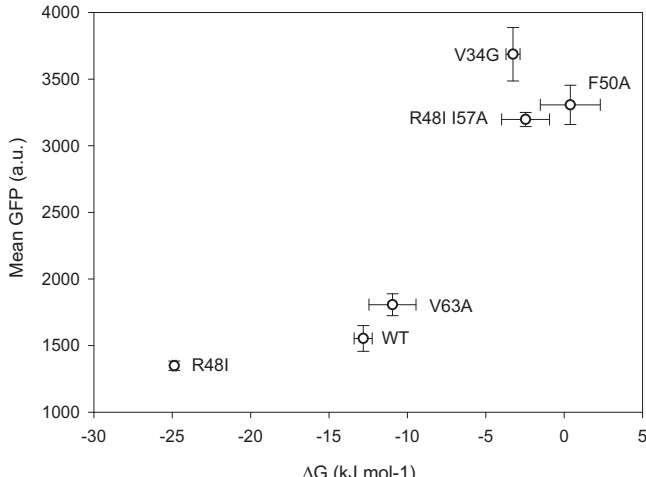

**Fig. 4 Correlation between GFP fluorescence and protein stability.** Six CI2 variants were co-expressed with the protein folding sensor (pSEVA631(Sp)-IbpAp-GFP-ASV), and the GFP fluorescence was analyzed by FACS. The average mean GFP fluorescence was compared to the Gibbs free energy of unfolding ($\Delta G_{unf}$) determined from global fits of thermal and chemical unfolding of each protein. All measurements were determined in triplicates and the data are presented as mean values ± standard deviation. Source data are provided as a Source Data file.

**Mutant library sequencing, stability analysis and decoy detection.** The PARP1-BRCT mutant library was sequenced by NGS both before and after sorting into the two populations (red (Lib. P1) and green (Lib. P2)) using FACS (Fig. 5c). For simplicity, we only investigated single site (amino acid) mutants. For a given mutant protein sequence, we compared its frequency in the green pool (destabilized proteins) with the frequency in the reference pool and used it as a proxy for protein stability. More specifically, for a given mutant protein sequence, we calculated the ratio between the high GFP fluorescence pool and the reference pool using Enrich2[43] that gives a score based on the normalized ratios. If the score is higher than 0, we consider the mutation to be neutral or stabilizing, and if the score was below 0, we consider it to be destabilizing (Fig. 5c bottom). As expected, full saturation mutagenesis was not obtained due to the low mutagenesis rate that makes it unlikely to have more than one nucleotide change per codon.

We then performed in silico calculations of the change in thermodynamic stability ($\Delta\Delta G$) of the BRCT domain using FoldX[44] and a solution NMR structure (PDB ID: 2COK) to assess how well predictions of thermodynamic stability correlate with the experimental data. We performed computational saturation mutagenesis in which we mutated each amino acid to all 19 other possible ones and calculated the change in stability (Fig. 5c bottom), and considered $\Delta\Delta G$ values ≥3 kcal/mol to be destabilizing (red x in Fig. 5c) and the remaining to be either neutral or stabilizing, to match the binary format of the sequencing data. Overall, we find a relatively low agreement between the FoldX calculations and the sequencing data, where destabilizing mutations based on the sequencing data (blue squares) are not always captured by FoldX. Enrich2 only ranks the observed mutations and does not classify the mutations as either stabilizing or destabilizing, thus changing the Enrich2 cut-off may either increase false positives or false negatives. To reduce possible noise, we analyzed the data position-wise by calculating the ratio between the number of destabilizing mutations and the number of total mutations for each position in the sequence ($N_{destab}/N_{total}$) for both FoldX and the experimental sequencing

data (Fig. 5c top). The ratio is high when most mutations result in destabilization and small when most mutations are neutral/stabilizing. From this analysis, we find a better correlation between the FoldX calculation and our experimental data (Fig. 5c top). Note that for the FoldX calculations we have performed full saturation mutagenesis, which means that $N_{total}$ is 19 for all positions, in contrast to the experimental data where $N_{total}$ varies. To remove the bias of selecting the $\Delta\Delta G$ cut-off for FoldX as well as summarizing across whole amino acid positions, we also performed a Receiver Operating Characteristic analysis (Fig. 5d). Here, the sequencing data provides the mutation specific labels (blue vs green in Fig. 5c) and the $\Delta\Delta G$s predicted from FoldX represent the predicted scores. From this analysis, we obtained an area under the curve of 0.61 suggestive of a reasonable but non-perfect correlation between the calculated stabilities and the experimental sequencing data.

Decoy detection in a protein structural ensemble is useful for protein structure prediction when using structural prediction tools such as Rosetta[45], which often produces a pool of candidate protein structures that might need additional filtering. Inspired by previous work that showed a correlation between the mutational tolerance of a site and how buried that site is in the protein structure[46], we examined whether the results from the folding sensor could be used in decoy discrimination. We used the mean of the Enrich2 scores for each position to individually score a pool of 20,000 structures generated by Rosetta. The assumption was that for a given residue in a native protein structural model, the residue depth should correlate with the mean Enrich2 score calculated from our experiments[46]. As an example, we depict a protein structure where each residue is colored either red or blue depending on their individual mean Enrich2 scores (Fig. 5e). Here, we find that low Enrich2 scores are likely attributed to residues in the core of the protein. Intuitively, one can imagine that the deeper a residue is embedded in the native protein structure, the more likely it is to destabilize the protein upon mutation due to packing issues. For each BRCT model we thus calculated the Spearman's correlation coefficient, ρ, to quantify the correlation between the residue depth and mutational tolerance (mean Enrich2 score). This correlation coefficient is considered as a structure specific score for which a higher coefficient is suggestive of a more native-like protein. To examine its usefulness in separating high quality structures from low quality structure, we plotted ρ as a function of the structural Global Distance Test – Total Score (GDT-TS) of the 20,000 generated structures with respect to the first conformer in the PDB structure (Fig. 5f). GDT-TS range from zero to one where one corresponds to a native or near native structure and zero is likely an extended protein. In Fig. 5f, we find a clear correlation between the structural scores ρ, and the structural quality defined by GDT-TS, suggesting that our experimental data can indeed be used to identify likely structures in a pool of candidate structures.

**Experimental validation of mutant variants identified through deep mutational scanning analysis.** From the deep mutational scanning we chose 20 variants for further analysis, of which 14 mutations (G20V, G20W, A31T, I33N, G37E, G37R, G37W, G38R, C50Y, S52I, S52N, I72N, V74F, H97L) were suggested by the deep mutational scan to result in misfolding and 6 mutations (I33T, V74I, D78V, Q81R, A96P, A96V) that were found not to interfere with protein folding. The 20 variants were synthesized and introduced into the translation sensor plasmid and analyzed using flow cytometry, and the protein concentration was quantified using Western blots (Fig. 6). The translation levels detected by the mCherry signal and by Western blotting was comparable for 6 (A31T, I33T, D78V, Q81R, A96P, A96V) of the 20 variants

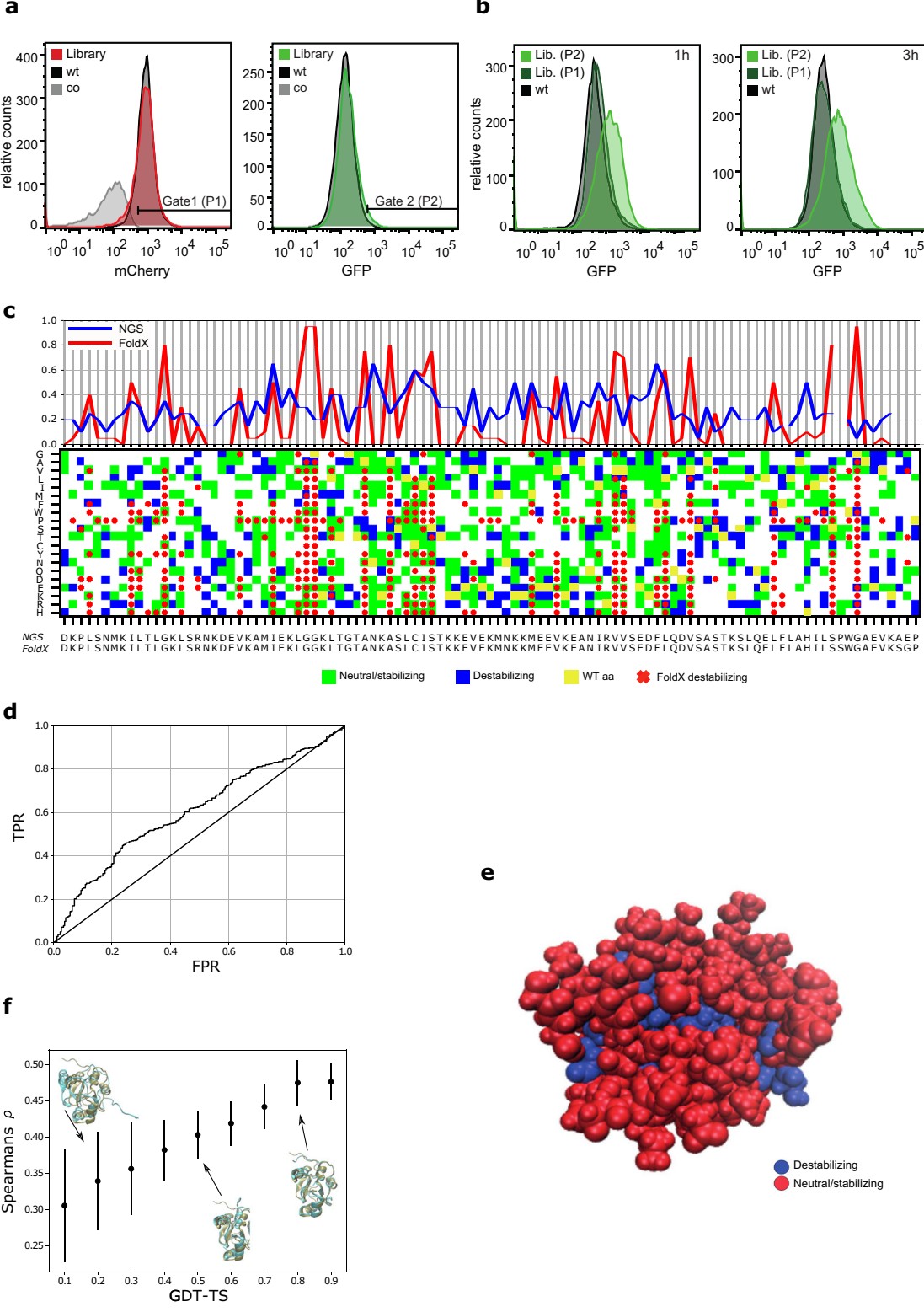

Neutral/stabilizing    Destabilizing    WT aa    FoldX destabilizing

(Fig. 6a). Translation levels of the remaining 14 variants, of which 13 were predicted to misfold, were detected by a mCherry signal, but with either no translation level detected by Western blot, or with Western blot signals distinctly lower than what would be expected based on the mCherry signal. This suggests that these variants were translated correctly but possibly degraded by proteases in the cell before detection by Western blot. Moreover, lower GFP signals in cells expressing variants with low or no

translation level detected by Western blot, suggests that the proteins were degraded before chaperones were able to bind and protect the unfolded or partially unfolded protein (Fig. 6b). Since the protein folding reporter is based on a chaperone promoter that is induced in the presence of misfolded protein, the system is capable of detecting misfolded proteins that would otherwise have been characterized as proteins with no expression. The PARP1-BRCT I33N and G37E variants were predicted to misfold, but in

**Fig. 5 FACS sorting and deep mutational scanning to identify variants of PARP1-BRTC with decreased protein folding. a** FACS sorting of PARP1-BRCT mutant library (red and green), PARP1-BRCT WT (black), and the translation sensor plasmid without a gene inserted (gray). Cells were sorted for high translation levels (Gate 1) and degree of protein misfolding (Gate 2). **b** The sorted cells were grown overnight and analyzed by flow cytometry 1 and 3 h after protein expression was induced. **c** Top: Ratio between the number of destabilizing mutations and the number of total mutations for each amino acid residue for both FoldX (red) and experimental data (blue). Bottom: Matrix plot indicating if an amino acid change (y-axis) of the sequence (x-axis) was destabilizing according to the high-throughput sequencing data as well as for FoldX calculations. For the experimental data, green and blue squares indicate neutral/stabilizing and destabilizing mutations, respectively. Yellow marks the wildtype to wildtype mutants, and white marks mutations with no experimental readout. Red x's indicate destabilizing mutations according to FoldX, with a cut-off of 3 kcal/mol. All squares without red x's are predicted to be neutral or stable mutants. **d** Receiver Operating Characteristic analysis of sequencing data and predicted FoldX ΔΔGs. The sequencing data provides the mutation specific labels (blue vs green in Fig. 5C) and the ΔΔGs predicted from FoldX are the mutation specific scores. **e** Structural visualization of stable vs destablilizing sequence positions of the PARP1-BRCT structure based on the experimental data. Blue residues that destabilize the protein have a $N_{destabl}/N_{total} \geq 0.2$, while the remaining are colored red. **f** Scoring of 20.000 structural decoys based on the experimental data. The plot shows the Spearman's correlation coefficient, ρ, that quantifies the correlation between residue depth and mutational tolerance based on the experimental data, as well as a structural quality measure defined by the structural Global Distance Test – Total Score (GDT-TS) score, where one corresponds to a native or near native structure. Here, the mean ρ is plotted for structures binned to the closest 0.1 GDT-TS bins. The error bars represent standard deviations for the individual bins. Source data are provided as a Source Data file.

contrast to the other variants, they were detectable by Western blotting, although still not at the same level as the mCherry signal, suggesting that there was a significant difference in the degradation rate of the mutants. The corresponding high GFP signal shows that the stability of the PARP1-BRCT I33N and G37E variants was decreased as also predicted from the library sequencing data and the FoldX analysis. For the variants predicted not to interfere with protein stability, there was a correlation between the level of protein measured by mCherry fluorescence and the amount of protein quantified by Western blotting. This observation was corroborated by the low GFP signals demonstrating that the mutations do not decrease the stability. The percentage of soluble protein was further quantified and visualized by Western blotting of the total protein fraction and the soluble protein fraction (Fig. 6c). As expected, variants with high GFP signals have low fractions of soluble protein, whereas variants with low or no GFP signal have high fractions of soluble protein.

**Identification of mutants with increased stability.** Having demonstrated the potential of the dual-reporter system for identifying residues important for protein folding, we expected that the reporter system can also be used to identify mutations that stabilize the folding of the protein. The PARP1-BRCT I33N variant was identified as a misfolded protein from the PARP1-BRCT mutant library, and we asked which variants, if any, might suppress the effect of I33N. PARP1-BRCT I33N was therefore used as a background for a new randomly generated mutant library with a mutation frequency of 1–3 mutations per protein, which was then co-expressed with the protein folding sensor pSEVA631(Sp)-IbpAp-GFP-ASV. Single cells that had high translation levels (Gate 1) as well as increased protein stability (Gate 2) were sorted by FACS (Fig. 7a, left panel). As expected PARP1-BRCT-I33N and the PARP1-BRCT-I33N mutant library resulted in significantly higher GFP signals than PARP1-BRCT-WT. After the first round of sorting, 64 single clones with high mCherry and low GFP signal (Gate 1 + Gate 2) were reanalyzed by flow cytometry, and one of the clones (1.5%) had a GFP signal overlapping with the GFP signal for PARP1-BRCT-WT) (Sort 1, Fig. 7b upper panel). After the second round of sorting, this population was further enriched to account for 12.5% of cells (Sort 2, Fig. 7b, lower panel)). A total of 64 clones were randomly selected from pool A and pool B and characterized by Sanger sequencing. Although the input library contained a wide range of mutations, all the selected clones were found to encode wild type PARP1-BRCT, except for one silent mutation, P10P, found after the second round of sorting. This mutation was caused by a

codon change from CCA to CCT, neither of which are characterized as rare codons[47]. These results demonstrate that it is possible to select more stable and correctly folded variants by successive rounds of FACS. Our observations that only WT sequences were found after sorting for proteins with increased stability is a likely result of the I33N mutation being a single nucleotide substitution making it likely to revert back to the WT PARP1-BRCT sequence. In addition, the most severely destabilizing single amino acid changes require multiple amino acid substitutions in order to recover or improve protein stability[15]. To increase the likelihood of finding more clones with increased stability, a library with a higher mutation frequency could be used. The results do show that it is possible to identify clones with decreased GFP signal from a mutant library, which in this case turned out to be the WT sequence.

## Discussion

The dual-reporter system presented here is a high-throughput screening method enabling fast and simultaneous monitoring of translation and protein folding at the single cell level. The setup has broad applicability and can be used as a screening tool to optimize expression conditions, testing different solubility and purification tags, as well as a tool for in deep mutational scanning and directed evolution studies. The folding-reporter plasmid can be used alone or together with the translational-coupling plasmid, in conjunction with expression of different pathway enzymes for synthetic biology trouble shooting. The use of the small hexahistidine tag for translational coupling to the reporter protein makes it possible to omit the use of a larger tag that may likely interfere with the three-dimensional structure of the protein and its function. Furthermore, the tag has the advantage that it can be used for downstream quantification and purification steps.

Formation of IB is often the bottleneck when expressing recombinant proteins in *E. coli*, and solubility tags or solubilization of the IB and subsequent refolding of the proteins are often necessary to recover folded and active proteins. In our reporter system, the GFP fluorescence is a result of the formation of aggregates and misfolded protein within the cell and is dependent on the presence of DnaK or DnaJ binding sites in the misfolded protein. The heat shock sigma factor, RpoH, needed for binding to the lbpA promoter on the protein folding sensor is released when DnaK binds the misfolded protein. The number of DnaK or DnaJ binding sites may influence the intensity of the GFP signal, as higher amounts of RpoH will be released with higher numbers of DnaK binding sites.

BRCA1-BRCT and E6 were insoluble when expressed in the reporter strain, however, with different levels of GFP

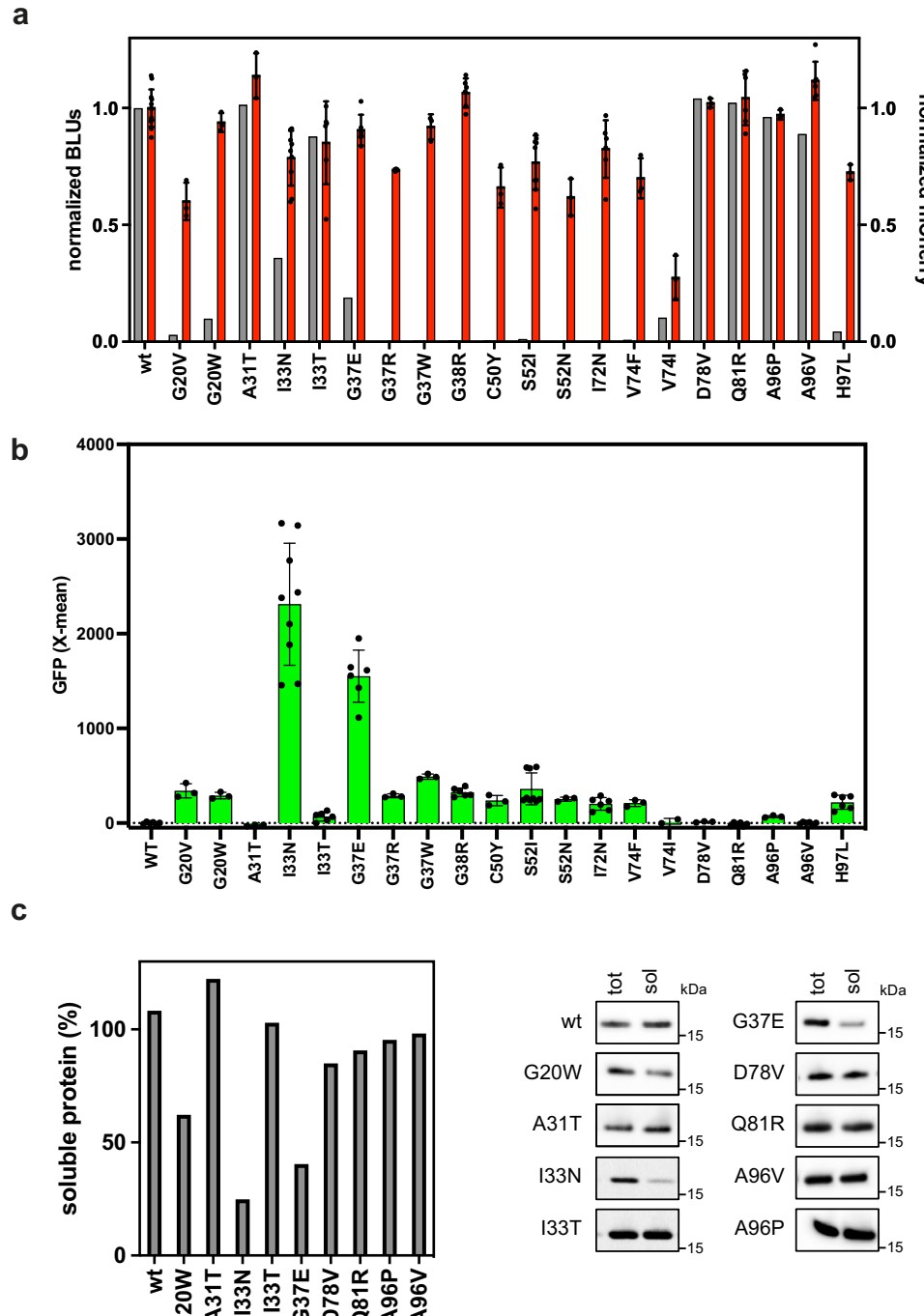

**Fig. 6 PARP1-BRCT mutants with changed folding properties identified from a randomly generated mutant library using the dual-reporter system.**
**a** Correlation between translation levels of 20 PARP1-BRCT mutants quantified from Western blots (gray, $n = 1$) and flow cytometry analysis of mean mCherry fluorescence values ± standard deviation (red), each normalized to the WT signal ($n \geq 3$, biologically independent samples). **b** GFP levels analyzed using flow cytometry as a measure for protein solubility and folding properties. Data are presented as mean values ± standard deviation for $n \geq 3$ biologically independent samples. **c** Percentage of soluble protein determined by Western blot for the 9 PARP1-BRCT mutants with a detectable GFP response signal. Western blot analysis of total protein yield (tot) and soluble protein (sol) after fractionated cell disruption ($n = 1$). Source data are provided as a Source Data file.

fluorescence. Using the Limbo DnaK binding site prediction tool[48], BRCA1-BRCT and E6 are predicted to contain four and two DnaK binding sites, respectively. Assuming that all DnaK binding sites are exposed in the unfolded protein, this may explain the difference in GFP intensity between the two proteins. This suggests that the GFP fluorescence may not be comparable when investigating unrelated proteins. For variants of the same

protein, the GFP output can be used as a direct measure of high or low protein stability, as we have demonstrated for CI2. The method is thus ideal when comparing the effect of different modifications to a protein. For any protein, the presence of a GFP signal strongly indicates that the target protein does not fold correctly. Even for systems with a lower dynamic range, it has previously been shown possible to identify stabilized protein

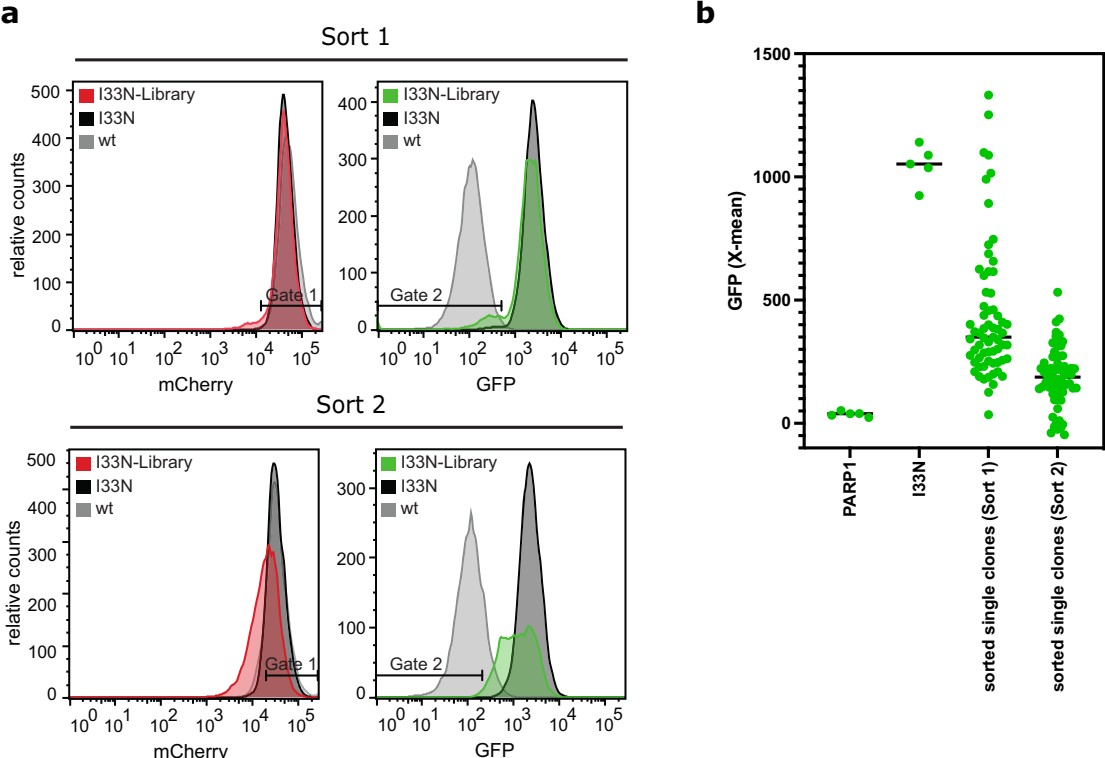

**Fig. 7 Identification of protein variants with improved folding properties from a PARP1-BRCT-I33N mutant library using the dual-reporter system.** A random PARP1-BRCT-I33N mutant library was co-expressed with the protein folding sensor. The cell populations were analyzed using FACS 1 h after IPTG induced protein expression. **a** FACS analysis of PARP1-BRCT WT, PARP1-BRCT-I33N, and the PARP1-BRCT-I33N mutant library, where the mCherry signal correlates with the translation level of PARP1-BRCT, while the GFP fluorescence is a measure of folding properties. Two gates were defined for sorting populations with high translation (Gate 1) and low GFP fluorescence (Gate 2), thus with improved folding properties compared to PARP1-BRCT-I33N. A shift is observed in GFP signal distribution and intensities between the two rounds of sorting, showing that it is possible to enrich the population with low GFP clones after multiple rounds of sorting. **b** Single clones analyzed after each round of sorting, resulted in 1.5% or 12.5% of the clones overlapping with the PARP1-BRCT WT GFP signal (n = 5 biologically independent samples for PARP1-BRCT WT, PARP1-BRCT-I33N and n = 1 for each of the 75 individual clones isolated from the library). Data are presented as individual data points with the mean value indicated. Source data are provided as a Source Data file.

variants[23,49]. In a recent study, the present reporter system has been used to successfully identify stabilized variants of the CI2 protein[50].

Mutant libraries are a main component of deep mutational scanning and directed evolution studies. The major drawback of randomly generated mutant libraries is the introduction of frame-shift mutations, stop-codons and indels that alters the amino acid sequence and results in nonfunctional proteins. The incorporation of the translation sensor makes it possible to differentiate between nonsense and missense mutations since the reporter signal will only be generated when complete translation of the target protein has been achieved.

Generation of mutant libraries using an error-prone DNA polymerase is limited by the genetic code, thus full saturation mutant libraries are difficult to obtain. Depending on the purpose of the experiment, the mutant libraries should be designed accordingly. Since multiple point mutations are often necessary for obtaining protein variants with improved overall stability, a mutant library for selecting stabilized variants should have a higher mutation frequency than a library for selecting destabilized variants. On the other hand, global analysis of libraries with different number of amino acid changes may provide detailed insight into protein folding and function[51–53].

We have shown that the mutational profile can be used to provide insight into the structure of a protein through decoy detection. Very recently it has been shown that more extensive deep mutational scans can be used to determine accurate three-

dimensional structures[54,55] and we envision that when the reporter system is used to select for stable protein variants it can be used in such structure-determination protocols.

Through screening for protein variants with improved stability from the destabilized PARP1-BRCT-I33N mutant, we successfully identified revertants to the wild type sequence, while a single silent mutation, PARP1-BRCT P10, was also identified. Where the amino acid sequence determines the three-dimensional fold of a protein the nucleotide sequence may affect the translational rate and thus co-translational folding of the proteins. Silent mutations may therefore still improve protein solubility and stability.

All destabilized variants found in the PARP1-BRCT library are situated in the core of the protein fold, which is consistent with the core being more sensitive to mutation, which often also results in loss of function[51,56]. When mutating enzymes for improved translation levels and protein folding it involves a risk of altering the activity of the enzyme. It is known that mutations within or close to the catalytic sites of enzymes may result in an improved stability of the protein, but with a corresponding decrease in enzyme activity[57,58]. The dual-reporter system can be used to obtain protein variants with high translation levels and high or moderate solubility; however, a downstream activity assay is needed to ensure an active enzyme. Activity assays are generally protein specific and are difficult to incorporate in a generalized high-throughput screening method. The reporter system can therefore be used to significantly reduce the number of variants that needs to be tested.

In summary, we have presented a dual-reporter biosensor system to assess in vivo protein translation and solubility with broad applicability and a reliable output. The reporter system can be used for a wide number of applications, including as a screening assay in directed evolution and deep mutational scanning studies to identify protein variants with high expression levels and improved protein stability in a high-throughput setup. The dual-reporter system is capable of identifying mutations that were not correctly predicted by computational tools, and we therefore envision that the experimental data that can be generated using the system may be valuable for further improving computational stability predictions.

## Methods

**Chemicals and enzymes**. Standard chemicals were purchased from Sigma Aldrich and sodium acetate was purchased from Scharlau, imidazole was purchased from PanReac AppliChem and IPTG was purchased from Fischer Bioreagents. Enzymes for standard cloning procedures were purchased from Thermo Fisher Scientific and New England Biolabs, respectively.

**Construction of a fluorescence-based protein folding reporter**. For construction of a protein folding sensor that reports on the formation of IB, the IbpA promoter (Genbank: LQ302077.1) from *E. coli* MG1655 was fused to either a stable (GFP-mut3; GenBank: LQ302079.1[11]) or a destabilized version of GFP (GFP-ASV; GenBank: LQ302078.1[11]). The GFP-ASV and GFP-mut3 were amplified by PCR using primer pairs and templates as indicated in Supplementary Table 1 and Supplementary Table 2. PCR products were cloned into pSEVA441 (GenBank: JX560339.1) using the *Xba*I and *Spe*I restriction sites, resulting in either pSEVA441-GFP-ASV or pSEVA441-GFP-mut3. The *E. coli* lbpA promoter was amplified by PCR (Supplementary Table 1) and cloned via the *Pac*I and *Xba*I restriction sites into pSEVA441-lbpAp-GFP-ASV and pSEVA441-lbpAp-GFP-mut3, respectively. To generate pSEVA631(Sp)-lbpAp-GFP-ASV or pSE-VA631(Sp)-lbpAp-GFP-mut3, the lbpAp-GFP reporter gene was subcloned via *Pac*I and *Spe*I into the pSEVA631 (GenBank: JX560348.1). Finally, the gentamicin cassette of pSEVA631 was replaced by the spectinomycin cassette of pSEVA441 using the *Spe*I and *Psh*AI restriction sites. All constructs were verified by Sanger sequencing.

**Fusion of proteins with a fluorescent translation-sensor**. A set of proteins were fused to the translation coupling cassette[30] (GenBank: LQ302080.1) followed by mCherry (GenBank: LQ302081.1). The BRCT-domain of human Poly [ADP-ribose] polymerase 1 (PARP1-BRCT, GenBank: LQ302082.1), a truncated version of BRCT-domain of human breast cancer 1, early onset (BRCA1-BRCT, GenBank: LQ302085.1 the human cyclin-dependent kinase 4 inhibitor D (p19, GenBank: LQ302086.1), and protein E6 from human *papillomavirus type 16* (GenBank: LQ302087.1) were amplified by PCR using the primers and templates as indicated in Supplementary Table 1. In addition, mCherry was amplified by PCR according to Supplementary Table 1. Each protein encoding DNA fragment was assembled with the mCherry-PCR fragment and *Nde*I and *Hin*dIII digested pET22b vector (Novagen), using a Gibson assembly reaction (New England Biolabs). The resulting expression vectors pET22b-XXX-trans-mCherry (XXX stands for the respective protein; see also Supplementary Table 2) comprise the coding sequence of the different proteins being linked via a C-terminal translation coupling cassette[30] to the ORF of mCherry. All cloned constructs were confirmed by Sanger sequencing.

**Cloning of NusA and SUMO fusion proteins**. For analyzing the impact of NusA and SUMO solubility protein-tags on expression and translation levels of either PARP1-BRCT, BRCA1-BRCT, p19, or E6, proteins were N-terminally fused to NusA (GenBank: LQ302088.1) and SUMO (GenBank: LQ302089.1), respectively[59,60]. Thereby, NusA and SUMO were amplified by PCR using the primers indicated in Supplementary Table 1 and inserted into pET22-XXX-trans-mCherry via the *Nde*I restriction site. The final expression reporter plasmids named pET22b-NusA-XXX-trans-mCherry and pET22b-SUMO-XXX-trans-mCherry (XXX stands for the respective protein; see also Supplementary Table 2), respectively, were all verified by sequencing.

**Impact of plasmid copy number and GFP stability on protein folding reporter assay sensitivity**. The impact of the vector copy number and intracellular turnover rate of GFP, respectively, on the protein folding reporter system was analyzed to optimize the readout sensitivity of the assay. Therefore, pSEVA631(Sp)-lbpAp-GFP-ASV and pSEVA631(Sp)-lbpAp-GFP-mut3 (pBBR1 origin), as well as pSEVA441-lbpAp-GFP-ASV and pSEVA441-lbpAp-GFP-ASV (ColE1 origin) (constructed as described above), were co-transformed with pET22b in *E. coli* Rosetta2™(DE3)pLysS (Novagen®). Transformants were selected on LB plates containing 25 μg/mL chloramphenicol, 50 μg/mL spectinomycin, and 100 μg/mL ampicillin. Single clones were inoculated in LB medium supplemented with the

corresponding antibiotics and grown at 37 °C and 300 rpm to an OD$_{600}$ of 0.5. IB formation in *E. coli* was induced by performing a heat-shock for 10 min at 42 °C. After heat shock, cells were grown for an additional 2.5 h at 37 °C and 300 rpm. Induction of the lbpAp promoter by IBs in single cells was monitored over time by changes of the GFP signal using flow cytometry (Instrument: BD FACS-Aria™SORP cell sorter; Laser 1: 488 nm: >50 mW, Filter: 505LP, 530/30-nm FITC, Laser 2: 561 nm: >50 mW; Filter: 600LP, 610/20-nm PE-Texas Red®). As control, the GFP signal in un-induced cells was monitored for each time point. The GFP (FITC-A, X-mean) values at each time point analyzed using the FlowJo V10 software were normalized to the corresponding background GFP signal.

To further investigate the impact of GFP stability on the sensitivity of the lbpAp-GFP reporter gene assay, pSEVA631(Sp)-lbpAp-GFP-ASV and pSEVA631(Sp)-lbpAp-GFP-mut3, respectively, were co-transformed with either pET22b, pET22-PARP1-BRCT-trans-mCherry or pET22-BRCA1-BRCT-trans-mCherry into *E. coli* Rosetta2™(DE3)pLysS (Novagen®). Transformants were selected on LB plates containing 25 μg/mL chloramphenicol, 50 μg/mL spectinomycin and 100 μg/mL ampicillin. Single clones were grown at 37 °C and 300 rpm in LB medium supplemented with the corresponding antibiotics. At OD$_{600}$ of 0.5–0.7 the expression of the human proteins was induced by addition of 0.5 mM IPTG. Directly after induction, the growth temperature was changed to 30 °C. Induction of the lbpAp-GFP variants by misfolded proteins was analyzed 1 h after induction using flow cytometry as mentioned above. For data analysis the GFP-signal (FITC-A, X-mean) was normalized to the respective GFP-signal of the vector control. To investigate the applicability of the system in other *E. coli* strains we co-transformed the pSEVA631(Sp)-lbpAp-GFP-ASV together with either pET22-PARP1-BRCT-trans-mCherry or pET22-BRCA1-BRCT-trans-mCherry into *E. coli* K-12 MG1655 (DE3). Transformants were selected on LB plates containing 50 μg/mL spectinomycin and 100 μg/mL ampicillin. Single clones were grown at 37 °C and 250 rpm in LB medium supplemented with the corresponding antibiotics. At OD$_{600}$ of 0.5–0.7 the expression of the human proteins was induced by addition of 0.5 mM IPTG. Directly after induction, the growth temperature was changed to 30 °C, and OD$_{600}$, the mCherry-signal (577,610), and GFP-signal (485,528) was analyzed in a fluorescent plate reader 1 and 3 h after induction.

**Determination of protein localization by fractionated cell disruption**. Intracellular localization of proteins was further analyzed by fractionated cell disruption. Here, cells (from 1 mL culture) were harvested either 1 h (for immunoblot analysis) or 3 h (for Instant Blue staining) after induction of protein expression. The cell pellet was resuspended in 50 μL resuspension buffer (20 mM Tris-HCl pH 7.5, 150 mM NaCl; 10 mM EDTA, 1 × HP-protease inhibitor mix (Serva)) and cells were broken by repeated cycles of freeze and thaw. Afterwards, cells were adjusted to a final OD$_{600}$ of 5 in resuspension buffer supplemented with benzonase (≥500 units; Sigma Aldrich). After 20 min incubation on ice, cells were spun down for 1 min at 500 × g to remove cell debris. The supernatant containing all soluble and insoluble proteins was transferred to a fresh reaction tube. An aliquot of the supernatant was taken, representing the total protein fraction (total). The remaining cell lysate was spun down for 15 min at 20,000 x g and the supernatant containing all soluble proteins was transferred into a new reaction tube (sol). The isolated fractions were separated on SDS-PAGE (RunBlue 4–20 %, Expedeon; NuPAGE®Bis-Tris gel 4–12%, Invitrogen) and analyzed by Instant Blue staining (Expedeon) and quantitative immunoblotting using an anti-His antibody (Novagen).

**Dual-reporter system for simultaneous monitoring of protein translation and folding in single *E. coli* cells**. To analyze the combined reporter system, pSE-VA631(Sp)-lbpAp-GFP-ASV and the protein expression reporter plasmids (pET22b-XXX-trans-mCherry, pET22b-NusA-XXX-trans-mCherry, pET22b-SUMO-XXX-trans-mCherry) were co-transformed into chemically competent *E. coli* Rosetta2™(DE3)pLysS (Novagen®). Transformants were selected on LB plates containing 25 μg/mL chloramphenicol, 50 μg/mL spectinomycin, and 100 μg/mL ampicillin. Single clones were grown in LB medium (supplemented with the corresponding antibiotics) at 37 °C and 300 rpm to an OD$_{600}$ of 0.5–0.7 and expression of proteins was induced by addition of 0.5 mM IPTG. Directly after induction, the growth temperature was changed to 30 °C. Protein expression and folding was analyzed 1 h after induction using flow cytometry as mentioned above. For data analysis, GFP (FITC-A, X-mean) signal was normalized to the corresponding PARP1-BRCT signal.

To confirm signal of the translation reporter, protein expression levels were further analyzed by instant blue staining and quantitative immunoblotting using an anti-His-Antibody. Cell-disruption was performed by freeze and thaw cycles as described before and the total protein fractions as well as intracellular localization of the proteins were analyzed. Western Blot signal was quantified using the Image J software[61].

**Identification of PARP1-BRCT mutants with altered folding properties using FACS**. To generate a PARP1-BRCT mutant library the PARP1-BRCT domain was randomly mutated, aiming at a mutation rate of 1–3 mutations per construct, using the GeneMorph II random mutagenesis kit (Agilent) according to manufacturer′s instructions. Primers and templates used for the reactions are indicated in

Supplementary Table 1. A megawhop reaction was performed with the random mutated PCR product as megaprimer and pET22-PARP1-BRCT-trans-mCherry as template. The resulting linear DNA fragments were transformed into MegaX DH10B™ T1R Electrocomp™ cells (Invitrogen) and transformants were selected on LB plates supplemented with 100 μg/mL ampicillin. The colonies (library size >100,000) were pooled and the plasmids were directly purified without further growth.

The vectors pET22b, pET22-PARP1-BRCT-trans-mCherry, and the created pET22-PARP1-BRCT-trans-mCherry mutant library were transformed into electro-competent Rosetta2(DE3)pLysS cells harbouring the protein folding sensor (pSEVA631(Sp)-IbpAp-GFP-ASV). After recovery, transformants were directly inoculated into 2 mL LB medium containing 20 μg/mL chloramphenicol, 50 μg/mL spectinomycin, 100 μg/mL ampicillin, and grown overnight at 37 °C and 300 rpm. Cells were transferred into fresh medium and grown at 37 °C and 300 rpm to an $OD_{600}$ of 0.5–0.7. Expression of proteins was induced by addition of 0.5 mM IPTG and the growth temperature of the culture was shifted to 30 °C. 1 h after induction, cells were analyzed by flow cytometry as mentioned above. 150,000 cells expressing a PARP1-BRCT mutant protein at wildtype level based on the translation sensor signal (Fig. 5A, gate 1), and which had an increased GFP signal (Fig. 5A, Gate 2) were sorted into 1 mL LB medium supplemented with antibiotics and grown overnight at 37 °C and 300 rpm. To further enrich the E. coli fraction harbouring proteins with altered folding properties, another round of protein expression and sorting (150,000 events) was carried out as described above.

The following day, the sorted cell population was again analyzed 1 h after induction of protein expression by flow cytometry. Subcellular localization of proteins in the sorted E. coli fraction was analyzed by Immunoblotting using an anti-His antibody as described above.

For NGS, plasmids were isolated from the sorted E. coli population. As control, plasmids were isolated from the PARP1-BRCT mutant library, which was used as starting material for sorting. Two 300 bp DNA fragments were amplified from the PARP1-BRCT library using a high-fidelity polymerase (primers as indicated in Supplementary Table 1). The amplified fragments were purified using AMPure XP beads (Beckman Coulter) to remove free primers and primer-dimer species. Both PCR-products were mixed in a one-to-one ratio.

Next, a PCR reaction was performed to attach Illumina sequencing adapters (Nextera XT Index Kit, Illumina) to the DNA fragments. For the reaction a KAPA HiFi HotStart Polymerase (Kapa Biosystems) was used. The resulting PCR products were purified with AMPure XP beads. The product size of the PCR reaction was verified on a Bioanalyzer DNA 1000 chip and the DNA was quantified using a Qubit® 2.0 Fluorometer. DNA fragments were normalized to 10 nM in 10 mM Tris pH8.5, 0.1% Tween 20. In order to reduce the background signal, the sample was spiked with 5% Phi-X control DNA (Illumina). The DNA was loaded onto the flow cell provided in the MiSeq Reagent kit v2, subjected to 300 cycles (Illumina), and sequenced on a MiSeq sequencing system (Illumina).

**Enrichment analysis**. The analysis was carried out using Enrich2 software[43]. However, due to issues running Enrich2 directly from raw fastq files, we converted the fastq files into Enrich2 compatible variant counts using python scripts. The scripts for doing this as well as an Enrich2 analysis config file are available at https://doi.org/10.11583/DTU.10265420. The script does the following: Reads were merged using FLASH v.1.2.11[62] and mapped to the reference sequence using bowtie2 v.2.3.4.1[63]. The SAM files that bowtie2 outputs are then parsed to create Enrich2 compatible variant count files.

**Folding properties of PARP1-BRCT single mutants**. To generate PARP1-BRCT single mutants, a two-fragment Gibson assembly reaction was performed. For each single mutant two overlapping DNA fragments were amplified by PCR using pET22b-PARP1-BRCT-trans-mCherry as template. Primer pairs are listed in Supplementary Table 3. Finally, the two DNA fragments were joined using Gibson Assembly® Cloning Kit (New England Biolabs) according to manufacturer's instructions. The sequence of each single mutant was confirmed by sequencing. Resulting mutant constructs are listed in Supplementary Table 3.

To examine the translation levels and protein stability of PARP1-BRCT single mutants, each mutant construct (Supplementary Table 3) was co-transformed with pSEVA631(Sp)-lbpAp-GFP-ASV into chemically competent E. coli Rosetta2™(DE3)pLysS (Novagen®). Protein expression was induced by addition of IPTG and protein translation and folding were analyzed by flow cytometry and quantitative immunoblotting as described before. To determine the percentage of soluble protein, the western blot signal was quantified using the Image J software.

**Isolation of PARP1-BRCT-I33N single mutants with rescued folding properties using the dual-reporter system**. A PARP1-BRCT-I33N library was generated as described before, using pET22b-PARP1-BRCT-I33N-trans-mCherry as template. The plasmids pET22b, pET22-PARP1-BRCT-I33N-trans-mCherry and the created pET22-PARP1-BRCT-I33N-trans-mCherry mutant library were transformed into electro-competent Rosetta2(DE3)pLysS cells harbouring the protein folding sensor (pSEVA631(Sp)-IbpAp-GFP-ASV). Protein expression was induced with IPTG and flow cytometry was performed as described above. 64 single clones that show protein expression (Fig. 7A, Pool 1,

Gate 1) in combination with a decreased GFP signal (Fig. 7A, Pool 1; Gate 2) were sorted in 200 μl LB medium supplemented with antibiotics and grown to stationary phase at 37 °C and 300 rpm. To further enrich the E. coli fraction harbouring proteins with rescued folding properties, a pool of 150,000 cells was sorted (identical gating as for single clones) into 1 mL LB medium supplemented with antibiotics and grown again overnight at 37 °C and 300 rpm. Subsequently, a second round of IPTG induction and sorting was performed to gain another 64 single clones (Fig. 7A, Pool 2; Gate 1 and 2). To verify GFP signal, all single clones (Pool 1 and Pool 2) were inoculated into fresh medium, protein expression was induced, and GFP expression was analyzed using a BD LSRFortessa™ cell analyzer in the HTS mode (Laser 1: 488 nm: >50 mW, Filter: 505LP, 530/30-nm FITC). Finally, plasmids were isolated from single clone cultures, which showed no GFP signal after induction, and analyzed by Sanger sequencing.

**FACS-based CI2 stability assay**. To generate five CI2 mutants with varying stabilities, Site-Directed II Lightning mutagenesis kit (Agilent Technologies) was used with CI2 WT as template. Each mutant was amplified by PCR using primer pairs as indicated in Supplementary Table 1. PCR products were cloned into pET22b-mCherry vector using the NdeI and SpeI restriction sites and joined using Gibson Assembly® Cloning Kit (New England Biolabs) according to manufacturer's instructions.

The CI2 variants were co-transformed with pSEVA631(Sp)-IbpAp-GFP-ASV into Rosetta2 (DE3) pLysS chemically competent cells and expressed in 50 ml LB media supplemented with 100 μg/μl ampicillin, 25 μg/ml chloramphenicol, and 50 μg/ml spectinomycin at 30 °C, 250 rpm to an $OD_{600}$ of 0.8. Protein expression was induced with 0.5 mM IPTG. Cells were extracted before and 1 h after induction and kept on ice until FACS analysis. The mCherry and GFP fluorescence was analyzed on a BD FACS-ARIA™SORP cell sorter as mentioned above.

**CI2 expression and purification for stability measurements**. CI2 variants transformed into Rosetta2 (DE3) pLysS competent cells and expressed in 1 L LB in the presence of 100 μg/μL ampicillin and 25 μg/ml chloramphenicol at 37 °C. Protein expression was induced at $OD_{600}{\sim}0.5$–0.7 with 0.5 mM IPTG and cells were further grown at 30 °C for 4–5 h. Cells were harvested by centrifugation at $5000 \times g$ for 20 min. Cell pellets were resuspended in 20 ml buffer A (20 mM sodium acetate pH 5.3) and frozen at −20 °C. Cell lysis was performed by two rounds of sonication (1 min, 80% amplitude, 0.5 cycles, (Hielscher UP200S)) followed by 30 min incubation on ice in presence of 1 mg DNase. Cell debris and protein aggregates were removed by centrifugation at $20,000 \times g$, 4 °C for 30 min. The supernatants were loaded onto a 1 mL HisTrap HP column (GE Healthcare) equilibrated with buffer A, and eluted with a gradient of buffer B (20 mM sodium acetate pH 5.3, 1 M imidazole) from 0 to 100 %. Fractions containing CI2 determined from SDS-PAGE analysis were concentrated and loaded onto a superdex75 10/300 GL column (GE Healthcare) equilibrated with 20 mM sodium phosphate pH 7.4, 150 mM NaCl. For buffer exchange the samples were concentrated and loaded onto a superdex75 10/300 GL column equilibrated with 50 mM MES pH 6.25. The purity of the proteins was assessed by SDS-PAGE and the protein concentration was determined using a spectrometer (PerkinElmer lambda40) with an extinction coefficient of 6990 $M^{-1}\,cm^{-1}$.

The CI2 variants were diluted to 10 μM in MES pH 6.25 with or without 6 M guanidium chloride. Using both solutions a dilution series of guanidium chloride ranging from 0 to 6 M guanidium chloride was prepared. Intrinsic tryptophan and tyrosine fluorescence of the CI2 variants was measured in triplicates using nanoDSF technologies on a Prometheus NT.48 instrument (nanoTemper technologies) with a temperature range from 15 to 95 °C with 1 °C/min increments. Global fitting of the temperature and denaturant unfolding was performed using the 330 and 360 nm fluorescence and $\Delta G_U$ and was obtained as described before[64].

**Generating and scoring decoy structures**. We generated 20,000 decoy structures using Rosetta's threading protocol[45] with PDBID: 2COK (Solution structure of BRCT domain of poly(ADP-ribose) polymerase-1) as a template. As a means to score a given decoy structure, we calculated the spearman's correlation coefficient ρ between the residue depths of the decoy structure and the mean Enrich2 positional score for the corresponding positions in the sequence.

**Reporting summary**. Further information on research design is available in the Nature Research Reporting Summary linked to this article.

## Data availability

Source data are provided with this paper in the supplementary document named "Source data". Additional data can be found at https://doi.org/10.11583/DTU.c.5633536.v1 [65]. Biological materials are available from the corresponding author upon request. Source data are provided with this paper.

## Code availability

The scripts for doing this as well as an Enrich2 analysis config file are available at https://doi.org/10.11583/DTU.c.5633536.v1.

# ARTICLE

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

## Acknowledgements

This work was supported by the Novo Nordisk Foundation through a grant to DTU Biosustain (NNF20CC0035580) as well as through grants for Protein OPtimization (POP) (NNF15OC0016360), a Hallas-Møller stipend (R173-A14446), and a project grant from the Lundbeck Foundation (R126-2012-12589). The pSEVA plasmids were a kind gift of Professor de Lorenzo, Centro Nacional de Biotecnologia-CSIC, Spain.

## Author contributions

A.Z., L.H., M.K., S.I.J., K.T., K.L.L., A.T.N. designed the experimental work, A.Z., L.H., A.K., M.K., S.I.J., A.T.N. performed the experiments, A.Z., E.P., L.H., L.E.P., M.K., K.T., K.L.L., A.T.N. analyzed the data, L.H., A.Z. wrote the paper, L.H., A.Z., M.K., K.T., K.L.L., A.T.N. edited and reviewed the paper, and all authors read and approved the final paper.

## Competing interests

A.T.N., A.Z. and R.L. have filed a provisional application on this work (EP3209795B1, US10544414B2). The application covers the use of the two-cassette reporter system for assessing gene target translation and folding. All other authors declare no competing interests.
