## [Peer Review File · Nature Communications]

Reviewers' Comments:

Reviewer #1:

Remarks to the Author:

Zutz et al introduce a new E.coli based dual-reporter system to simultaneously quantify translation and folding levels. The system is based on the innovative use of fluorescent probes that are activated 1. during transcription and 2. if the protein goes into inclusion bodies. This enables the high-throughput selection of well-transcribed and expressed proteins. The field of high-throughput selection for expression is quite mature, but the use of these specific tags is not known to me and is interesting. Although the experimental setup is innovative, there are significant flaws in the experimental design that reduce the relevance and usefulness of the results as explained below:

Major concerns:

1. The most significant experimental-design flaw is the use of a marker for the formation of inclusion bodies and using this marker as a proxy for protein-expression levels. First, E.coli protein expression levels are due, not only to the formation of inclusion bodies but also due to proteolysis and there's literature spanning some 20 years that shows that bacterial proteases break down proteins with some correlation to their instability. Hence, the most unstable proteins would not go into inclusion bodies, simply because they would be proteolyzed. Therefore, it is not clear to me what advantage is offered by the marker that reports on inclusion bodies over standard GFP-tagged proteins.

2. The selections reported towards the end of the Results section show that no stabilized variants could be selected from this library. The authors nevertheless end on an optimistic note, saying that "the results show that it is possible to enrich the population...and thus select for improved stability". I think that without actual data on such increases, this optimism would not be shared by the readers.

3. As the authors note (line 393), the GFP fluorescence levels may not be comparable for different proteins. That's very likely to be true, meaning that the system would need to be calibrated for each protein under study. It's quite a laborious system as it is, suggesting that it would not be widely adopted.

4. Perhaps most importantly, this selection method does not address in any way protein activity. The proteins are allowed to mutate and mutants are selected simply based on transcription levels and on low levels of inclusion bodies. Given the well-known tradeoffs between activity and stability in many proteins, many of the most frequent mutations that come out of such a screen would disable the protein's activity. In most scenarios, harming activity is simply not acceptable and it is difficult for me to imagine a scenario where this system would be useful.

Reviewer #2:

Remarks to the Author:

In this paper the authors present a simple two-plasmid system that allows for the monitoring of protein expression and stability using fluorescence in E. coli cells. Protein expression is monitored by co-expression of an mCherry reporter through an operon structure in the transcriptional unit expressing the protein of interest. In parallel, expression of GFP is driven by the endogenous *ibpA* promoter. The major advantage of using a two-colour reporter is that it enables single cell analysis and sorting, opening up opportunities to explore pooled mutant libraries in high throughput. The authors demonstrate these capabilities with a number of examples using previously studied proteins.

Overall, the paper is well-written, clear, and the methods and data appear to be sound. I liked the work and believe that the tool is useful for some specific cases but worry that the target audience might be too narrow for Nature Communications. Also, while the experiments are more thorough than other studies, I worry that the system is so specific to E. coli, and even the strain of E. coli used in the work, that the results may not translate as broadly as the authors would hope. This is crucial to ensure the method is to have long-term and lasting impact. I also had the following

more specific comments:

Line 41: "to improve proteins for efficient expression". It is not clear from this what precisely you mean. Are you referring to the protein amino acid sequence? The expression system (e.g. promoter, RBS, CDS sequence, etc)? You go on later to explain the different aspects, but here be clear what you mean by improve.

Line 53: You have a very strange choice of paper for when referring to co-translational folding, why not one that actually shows its role: e.g. Zhang et al. "Transient ribosomal attenuation coordinates protein synthesis and co-translational folding" *Nature Chemical and Molecular Biology* 16, 274-280 (2009). Please also check through your other references that they touch upon relevant primary research literature, not broad unfocused reviews.

Line 55: The view painted is one where there are lots of factors that can be tuned, but what is missing is that many of these factors are unavoidably linked too, e.g., see: Gorochoowski et al. "Trade-offs between tRNA abundance and mRNA secondary structure support smoothing of translation elongation rate" *Nucleic Acids Research* 43, 2993-3011 (2015), and Cambray et al. "Evaluation of 244,000 synthetic sequences reveals design principles to optimize translation in *Escherichia coli*" *Nature Biotechnology* 36, 1005-1015 (2018). I would consider briefly raising this difficulty to the reader so that they understand the difficulties in this area.

Line 81: "controlling and aiding the process, and preventing unproductive misfolding." should read "controlling and aiding the process to prevent unproductive misfolding."

Line 84: "misfolded protein is" should read "misfolded proteins are"

Line 118: "mCherry being expressed in a one to one ratio with the protein of interest." This can be misleading. While I agree that the translation rate will have a near one-to-one ratio, the expression level will most certainly not as it is unlikely both proteins have identical degradation rates. It would be correct to say that the expression of one is proportional to the other.

Line 122: "other form of stress conditions" should read "other forms of stress condition"

Line 127: "RpoH then binds to the RNA polymerase sigma70, which subsequently recognizes heat shock promoters and thus initiate a heat shock response." This is not correct. RpoH does not bind sigma70. RpoH has a higher affinity to the RNAP core enzyme than sigma70 at higher temperatures and so displaces it, activating "heat-shock" regulated genes. Given the central use of this system, please ensure you describe it correctly using the best scientific knowledge we have.

Line 157 "The highly stable GFP-mut3 slowly accumulates in the cell over time and results in higher GFP signals for both heat induced pBBR1 and the control, thereby resulting in a lower signal-to-noise ratio". This is not strictly correct. It would be better to say that the more stable GFP has a higher basal expression level at steady state than the destabilised one, causing a greater overlap in the distributions and lower signal to noise ratio.

Line 182: "aggregates" should read "aggregate"

Line 218: "protein folding sensor can be used as a proxy for the in vitro stability of variants" This idea is suggested in numerous places throughout the paper and while I agree the sensor is able to detect highly unstable proteins, I am less convinced and see no strong evidence in this data that stability and the sensor are highly correlated. Even in Figure 4 which is used to substantiate this claim, the data displays more of a switch, not a graduated expression in response to stability. It might be possible to elucidate this information by varying the expression level of the protein such that even a mildly unstable protein at very high concentrations might sufficiently trigger a response, and looking at a combination of promoter strength (not merely inducer concentration) when this happens as a more linear stability measure. I would recommend, either performing more experiments to demonstrate this claim properly (with intermittent stability), or to clarify that the sensor is more binary (stable/unstable) in its response.

Line 278: "and do not" should read "and does not"

Line 344: I found it disappointing that a reversion to WT was the stabilising mutation found. Would there be ways to guide/bias the mutational landscape to provide a higher likelihood of hitting on a compensatory mutation (e.g. could molecular modelling help)?

Line 429: I do not understand why there is a separate Conclusion section that repeats a lot from the Discussion. I would recommend merging these together keeping the conclusions concise.

We would like to thank the reviewers for their careful reviews and their comments. We have replied to all comments and also modified the manuscript as detailed in red below. We believe that this has significantly improved the manuscript.

Reviewers' comments:

Reviewer #1 (Remarks to the Author):

Zutz et al introduce a new E.coli based dual-reporter system to simultaneously quantify translation and folding levels. The system is based on the innovative use of fluorescent probes that are activated 1. during transcription and 2. if the protein goes into inclusion bodies. This enables the high-throughput selection of well-transcribed and expressed proteins. The field of high-throughput selection for expression is quite mature, but the use of these specific tags is not known to me and is interesting. Although the experimental setup is innovative, there are significant flaws in the experimental design that reduce the relevance and usefulness of the results as explained below:

Major concerns:

1. The most significant experimental-design flaw is the use of a marker for the formation of inclusion bodies and using this marker as a proxy for protein-expression levels. First, E.coli protein expression levels are due, not only to the formation of inclusion bodies but also due to proteolysis and there's literature spanning some 20 years that shows that bacterial proteases break down proteins with some correlation to their instability. Hence, the most unstable proteins would not go into inclusion bodies, simply because they would be proteolyzed. Therefore, it is not clear to me what advantage is offered by the marker that reports on inclusion bodies over standard GFP-tagged proteins.

Reply: Thank you for the comment. The reporter system does actually not sense the presence of inclusion bodies – just the presence of misfolded protein in the cell. We have tried to explain this in detail in line 125-138. In the revised version, we have removed the mentioning of inclusion bodies in line 137 as this could be misleading. In the example shown in figure 6a and 6b, it is also clear that the system correctly detects both the expression and misfolding of mutant variants that cause the protein to be degraded and not accumulate in inclusion bodies.

When using a more traditional tagging of a target protein with a fluorescent protein there is a risk that the fusion itself can interfere with the folding and thereby give misleading results. Furthermore, there is a risk that the GFP signal will be degraded if the target protein is targeted for proteolysis. The present reporter system is generic and works for all proteins, and it does not depend on protein fusions.

2. The selections reported towards the end of the Results section show that no stabilized variants could be selected from this library. The authors nevertheless end on an optimistic note, saying that "the results show that it is possible to enrich the population...and thus select for improved stability". I think that without actual data on such increases, this optimism would not be shared by the readers.

Reply: We do agree that with the reviewer that it would have been more interesting to find alternative compensatory mutations. However, starting from the I33N mutant, which does not fold correctly, we were able to create a random mutant library and from this to identify variants that had correct folding. These just happened to be the wild type sequence. We have tried to modify the sentence starting in line 378 so that it now reads: "The results do show that it is possible to identify clones with decreased GFP signal from a mutant library, which in this case turned out to be the WT sequence."

We have recently carried out a very extensive independent study of the Cl2 protein, where it was indeed possible to identify stabilizing mutations. In addition, we were able to use the reporter system to study synergistic point mutations. We are therefore convinced that this approach is working.

3. As the authors note (line 393), the GFP fluorescence levels may not be comparable for different proteins. That's very likely to be true, meaning that the system would need to be calibrated for each protein under study. It's quite a laborious system as it is, suggesting that it would not be widely adopted.

Reply: It is true that we observe a difference in GFP level for different misfolded proteins. However, the system can still readily be used to identify if there is misfolded protein present in the cell if a GFP signal is present. We expect that the difference in GFP signal for different misfolded proteins is caused by differences in binding of DnaK to the proteins. For mutant variants of the same protein, the GFP fluorescence level only varies as a function of folding. As shown, the system can be used with plate readers or by FACS analysis, where the calibration is simply done by adjusting instrument settings. We have tried to explain this better in the discussion, line 405: "This suggests that the GFP fluorescence may not be comparable when investigating unrelated proteins. For variants of the same protein, the GFP output can be used as a direct measure of high or low protein stability, as we have demonstrated for Cl2. The method is thus ideal when comparing the effect of different modifications to a protein. For any protein, the presence of a GFP signal strongly indicates that the target protein does not fold correctly."

4. Perhaps most importantly, this selection method does not address in any way protein activity. The proteins are allowed to mutate and mutants are selected simply based on transcription levels and on low levels of inclusion bodies. Given the well-known tradeoffs between activity and stability in many proteins, many of the most frequent mutations that come out of such a screen would disable the protein's activity. In most scenarios, harming activity is simply not acceptable and it is difficult for me to imagine a scenario where this system would be useful.

Reply: Thanks for raising this point, which we also tried to address at the end of the discussion. Activity assays are generally difficult to incorporate for high-throughput screening, unless the product is fluorescent or can be coupled to fitness of the cell. It will therefore be an advantage to use this reporter system as a high throughput pre-screen, which will help reduce the number of variants that needs to be tested using an activity assay. We have tried to make this clearer in the discussion starting from line 439: "The dual-reporter system can be used to obtain protein variants with high translation levels and high or moderate solubility; however, a downstream activity assay is needed to ensure an active enzyme. Activity assays are generally protein specific and are difficult to incorporate in a generalized high-throughput screening method. The reporter system can therefore be used to significantly reduce the number of variants that needs to be tested."

The developed system may of course also be used for various other experiments, for example deep mutational scanning with the purpose of generating information for better computational prediction of protein folding or stability. The reporter system may also be valuable for production of biochemicals using metabolic engineering. Here, multiple pathway enzymes from different organisms are typically expressed, and the system may be used to quickly test if these pathway enzymes are folded correctly.

Reviewer #2 (Remarks to the Author):

In this paper the authors present a simple two-plasmid system that allows for the monitoring of protein expression and stability using fluorescence in E. coli cells. Protein expression is monitored by

co-expression of an mCherry reporter through an operon structure in the transcriptional unit expressing the protein of interest. In parallel, expression of GFP is driven by the endogenous *ibpA* promoter. The major advantage of using a two-colour reporter is that it enables single cell analysis and sorting, opening up opportunities to explore pooled mutant libraries in high throughput. The authors demonstrate these capabilities with a number of examples using previously studied proteins.

Overall, the paper is well-written, clear, and the methods and data appear to be sound. I liked the work and believe that the tool is useful for some specific cases but worry that the target audience might be too narrow for Nature Communications. Also, while the experiments are more thorough than other studies, I worry that the system is so specific to *E. coli*, and even the strain of *E. coli* used in the work, that the results may not translate as broadly as the authors would hope. This is crucial to ensure the method is to have long-term and lasting impact. I also had the following more specific comments:

Reply: We would like to thank the reviewer for his encouraging comments. To demonstrate that the system is not only limited to the *E. coli* Rosetta2 strain, we have now included an experiment that shows that the system also works very well in *E. coli* K-12 MG1655 (DE3) (Figure 2e and described from line 180). It is expected that the system will work in other *E. coli* strains as well. The Rosetta2 strain is derived from the BL21 strain, and together with MG1655, these are properly the two most widely used strains for both protein production and also metabolic pathway engineering. *E. coli* is a preferred microorganism for expressing therapeutic proteins, and around 30% of the approved therapeutic proteins are currently being produced in *E. coli*. We therefore believe that the reporter system will find a lot of use. Similar reporter systems could be generated for other organisms, including gram positive bacteria, if a suitable chaperone promoter is identified.

Line 41: “to improve proteins for efficient expression”. It is not clear from this what precisely you mean. Are you referring to the protein amino acid sequence? The expression system (e.g. promoter, RBS, CDS sequence, etc)? You go on later to explain the different aspects, but here be clear what you mean by improve.

Reply: Thank for pointing this out. We agree that it was not clear and have modified the sentence so that it now reads: “It is therefore of significant importance to be able to efficiently modify the protein coding sequence in a way that will enable more efficient folding and expression.”

Line 53: You have a very strange choice of paper for when referring to co-translational folding, why not one that actually shows its role: e.g. Zhang et al. “Transient ribosomal attenuation coordinates protein synthesis and co-translational folding” Nature Chemical and Molecular Biology 16, 274-280 (2009). Please also check through your other references that they touch upon relevant primary research literature, not broad unfocused reviews.

Reply: Thanks for pointing this out – we have now corrected the specific reference and also gone through the manuscript to check for more relevant references.

Line 55: The view painted is one where there are lots of factors that can be tuned, but what is missing is that many of these factors are unavoidably linked too, e.g., see: Gorochofski et al. “Trade-offs between tRNA abundance and mRNA secondary structure support smoothing of translation elongation rate” Nucleic Acids Research 43, 2993-3011 (2015), and Cambrey et al. “Evaluation of 244,000 synthetic sequences reveals design principles to optimize translation in Escherichia coli” Nature Biotechnology 36, 1005-1015 (2018). I would consider briefly raising this difficulty to the reader so that they understand the difficulties in this area.

Reply: This is a very good point. We have added a sentence at the end of the paragraph (line 66) addressing this: “Changing only one variable may not have the desired effect as many of these factors are linked and have a synergistic effect, thus emphasizing that the optimization process is not a straightforward task (10,11)”.

Line 81: “controlling and aiding the process, and preventing unproductive misfolding.” should read “controlling and aiding the process to prevent unproductive misfolding.”

Reply: Thanks for catching this – we have corrected the sentence as suggested.

Line 84: “misfolded protein is” should read “misfolded proteins are”

Reply: Thanks for catching this – we have corrected the sentence as suggested.

Line 118: “mCherry being expressed in a one to one ratio with the protein of interest.” This can be misleading. While I agree that the translation rate will have a near one-to-one ratio, the expression level will most certainly not as it is unlikely both proteins have identical degradation rates. It would be correct to say that the expression of one is proportional to the other.

Reply: This is also a good point – we have changed the sentence so that it now reads: “Correct translation of the gene of interest results in the expression of mCherry being proportional to the expression of the protein of interest.”

Line 122: “other form of stress conditions” should read “other forms of stress condition”

Reply: Thanks for catching this – we have corrected the sentence as suggested.

Line 127: “RpoH then binds to the RNA polymerase sigma70, which subsequently recognizes heat shock promoters and thus initiate a heat shock response.” This is not correct. RpoH does not bind sigma70. RpoH has a higher affinity to the RNAP core enzyme than sigma70 at higher temperatures and so displaces it, activating “heat-shock” regulated genes. Given the central use of this system, please ensure you describe it correctly using the best scientific knowledge we have.

Reply: Thanks for pointing this out. We have now changed the sentence so that it reads: “RpoH then binds to the core RNA polymerase forming a holoenzyme complex, which subsequently recognizes heat shock promoters and thus initiate a heat shock response.”

Line 157 “The highly stable GFP-mut3 slowly accumulates in the cell over time and results in higher GFP signals for both heat induced pBBR1 and the control, thereby resulting in a lower signal-to-noise ratio”. This is not strictly correct. It would be better to say that the more stable GFP has a higher basal expression level at steady state than the destabilised one, causing a greater overlap in the distributions and lower signal to noise ratio.

Reply: This is an interesting point, and we agree that it could be described better. The ASV tag does not change the expression (rate) – only the half-life of the protein. We have tried to modify the sentence to make it clearer: “The highly stable GFP-mut3 had a high basal fluorescence level causing a significant overlap between the induced and control responses resulting in a low signal-to-noise ratio. The use of the GFP-ASV variant with shorter half-life resulted in a lower basal fluorescence level thereby giving higher signal-to-noise ratios, which enabled the distinction of the heat shock induced response from protein misfolding in single cells.”

Line 182: “aggregates” should read “aggregate”

Reply: Thanks for catching this – we have corrected the sentence as suggested.

Line 218: “protein folding sensor can be used as a proxy for the in vitro stability of variants” This idea is suggested in numerous places throughout the paper and while I agree the sensor is able to detect highly unstable proteins, I am less convinced and see no strong evidence in this data that stability and the sensor are highly correlated. Even in Figure 4 which is used to substantiate this claim, the data displays more of a switch, not a graduated expression in response to stability. It might be possible to elucidate this information by varying the expression level of the protein such that even a mildly unstable protein at very high concentrations might sufficiently trigger a response, and looking at a combination of promoter strength (not merely inducer concentration) when this happens as a more linear stability measure. I would recommend, either performing more experiments to demonstrate this claim properly (with intermittent stability), or to clarify that the sensor is more binary (stable/unstable) in its response.

Reply: We agree with the reviewer that there is not a direct correlation between the stability and the GFP fluorescence level, however, it is possible to select variants with increased or decreased stability when compared to the wild type sequence. We have modified the sentence so that it now reads: “These results show that the GFP fluorescence arising from the protein folding sensor can be used as a proxy for the in vitro stability of variants in a mutant library, by characterizing the stability for each variant as higher or lower than the starting point.”

Line 278: “and do not” should read “and does not”

Reply: Thanks – we have corrected the sentence as suggested.

Line 344: I found it disappointing that a reversion to WT was the stabilising mutation found. Would there be ways to guide/bias the mutational landscape to provide a higher likelihood of hitting on a compensatory mutation (e.g. could molecular modelling help)?

Reply: We can only agree with the reviewer’s comment. We also were hoping for to identify other compensatory mutations. However, with the given starting point, it may be expected that the only the wild type sequence will restore correct folding. It would certainly be possible to test this out with mutations that require more than one nucleotide change to restore to the wild type sequence. Given the high throughput of the method, it is feasible to screen complete random mutant libraries. However, the approach could just as well be used for screening more focused libraries based on computational approaches, which could for example include combinations of mutations that were predicted to be stabilizing.

In a separate extensive study, we have recently demonstrated how it is indeed possible to identify other stabilizing mutations in the Chymotrypsin inhibitor 2 (CI2). We also studied how point mutations may act synergistically to stabilize a protein structure. We considered merging the two manuscripts, but the work is too extensive to fit into a single publication.

Line 429: I do not understand why there is a separate Conclusion section that repeats a lot from the Discussion. I would recommend merging these together keeping the conclusions concise.

Reply: We do agree with this and have now largely removed the conclusion section.

Reviewers' Comments:

Reviewer #1:

Remarks to the Author:

The authors' revision certainly addresses the points raised during the review. Tools, however, are only as good as they are useful and the evidence presented in this paper that the method can detect stabilising mutants is a reversion of a single-point mutant to wild type. In their letter, the authors state that they've used the system to screen large pools of mutants and successfully isolated stabilising ones. This is very encouraging, but I strongly recommend that these new data and analysis be presented together with the current submission. The current submission's usefulness is not convincing without those data.

Reviewer #2:

Remarks to the Author:

Overall, the authors have addressed most of my previous queries. I still have some reservations as to the generality of the approach to other organisms but concede that the authors have at least shown the system works for commonly used protein production strains of *E. coli*. However, there is one major outstanding point that was not appropriately addressed in the revision:

1. I still contest that Figure 4 does not support the fact that GFP fluorescence can be used as an accurate proxy for stability. The correction made to the text still states something that is not backed up by data. For example, to me it looks like GFP expression is either low or high (binary) and there is no evidence that GFP can tell you anything more than stability being low or high – which is not that useful in this situation. I'd suggest the authors carefully consider whether it is possible to perform experiments that can clearly show that a true correlation is present, or more accurately describe the relationship seen in the data they have and the limitations this imposes on their methodology.

REVIEWER COMMENTS

Reviewer #1 (Remarks to the Author):

The authors' revision certainly addresses the points raised during the review. Tools, however, are only as good as they are useful and the evidence presented in this paper that the method can detect stabilising mutants is a reversion of a single-point mutant to wild type. In their letter, the authors state that they've used the system to screen large pools of mutants and successfully isolated stabilising ones. This is very encouraging, but I strongly recommend that these new data and analysis be presented together with the current submission. The current submission's usefulness is not convincing without those data.

Reply: Thanks for your positive comments. The reason that we hesitated to include the demonstration of the additional application in the present manuscript is that the work was a very extensive and would have been difficult to include. Therefore, with support from the editor, the second manuscript was submitted to *Communication Biology*, where it has received very positive reviews. A Final Revision of the manuscript has now been resubmitted with minor editorial changes. A recent version of the manuscript is available for your information here:

<https://www.biorxiv.org/content/10.1101/2020.12.01.406082v1>

The manuscript has now been cited in the present manuscript:

Line 411: "In a recent study, the present reporter system has been used to successfully identify stabilized variants of the Cl2 protein⁵⁰."

We hope that you will agree, when looking at the manuscript, that it would be difficult to merge it into the present paper, and that it serves as an additional demonstration of the application of the present tool for investigating protein translation and folding.

Reviewer #2 (Remarks to the Author):

Overall, the authors have addressed most of my previous queries. I still have some reservations as to the generality of the approach to other organisms but concede that the authors have at least shown the system works for commonly used protein production strains of *E. coli*. However, there is one major outstanding point that was not appropriately addressed in the revision:

1. I still contest that Figure 4 does not support the fact that GFP fluorescence can be used as an accurate proxy for stability. The correction made to the text still states something that is not backed up by data. For example, to me it looks like GFP expression is either low or high (binary) and there is no evidence that GFP can tell you anything more than stability being low or high – which is not that useful in this situation. I'd suggest the authors carefully consider whether it is possible to perform experiments that can clearly show that a true correlation is present, or more accurately describe the relationship seen in the data they have and the limitations this imposes on their methodology.

Thanks for your comments. We do agree that the data presented in figure 4 is more of a binary nature. We have therefore modified the text to better reflect this:

- Line 27: The abstract has been slightly modified by leaving out the word “intensity”: “We have validated the dual-reporter system on five different proteins and find an excellent correlation between reporter signals and the levels of protein expression and solubility of the proteins.”
- Line 98: The sentence has been modified so that it now only mentions that the tool can be used to identify the occurrence of protein misfolding: “Here, we demonstrate a functional dual reporter system that enables single-cell monitoring of both protein translation levels and the occurrence of protein misfolding.”
- Line 222: The sentence has been modified to say that more unstable proteins result in high signals: “The GFP fluorescence clearly changed with ΔG_U , where more unstable proteins resulted in high GFP signals (Figure 4).”

It has previously been shown that it is possible to use reporter systems with a lower dynamic range for identifying mutations that stabilize proteins. We have now mentioned and cited this in the manuscript. We have also introduced a citation to a manuscript mentioned above (see the reply to reviewer 1), where we demonstrate the use of the reporter system for identifying stabilizing variants of the Cl2 protein:

- Line 409: “Even for systems with a lower dynamic range, it has previously been shown possible to identify stabilized protein variants^{23,49}. In a recent study, it the reporter system has been used to successfully identify stabilized variants of the Cl2 protein⁵⁰.”

We hope that these modifications to the manuscript will better reflect the data presented in Figure 4.

Reviewers' Comments:

Reviewer #2:

Remarks to the Author:

I am happy that the authors have addressed my previous concerns appropriately.